# Structural interactions of BWC0977 with *Klebsiella pneumoniae* topoisomerase IV and biochemical basis of its broad-spectrum activity

Radha Nandishaiah [1,2], Satoshi Murakami [3] ✉, Shahul Hameed P[1], Maho Aoki [3], Ui Okada [3], Eiki Yamashita [4], Suryanarayanan Venkatesan[1], Nagakumar Bharatham[1], Sudipta Sarma[1], Anirudh P. Shanbhag [1], Sreevalli Sharma [1], Ranga Rao[1], Vasanthi Ramachandran [1], V. Balasubramanian [1], Santanu Datta [1] & Nainesh Katagihallimath [1] ✉

Antimicrobial resistance is a growing global health crisis driving the urgent need for effective broad-spectrum antibiotics. BWC0977 is a pyrazino-oxazinone based novel bacterial topoisomerase inhibitor (NBTI) currently in Phase 1 clinical trials and demonstrates potent activity against multidrug-resistant Gram-negative and Gram-positive bacteria. It targets both DNA gyrase and topoisomerase IV with balanced low-nanomolar potencies, showing remarkable superiority over ciprofloxacin and gepotidacin. We report the first 3.05 Å cocrystal structure of BWC0977 bound to *Klebsiella pneumoniae* topoisomerase IV, revealing its binding mode and interaction residues. The reduced inhibition of BWC0977 against purified gyrase enzymes carrying an individual mutation at these residues supports the relevance of these molecular interactions. Mutational analyses in *Escherichia coli* strains show that single target mutations in *gyrA* or *parC* do not confer resistance, while simultaneous mutations in both genes result in over 250-fold reduced susceptibility. The compound also demonstrates more than 5000-fold selectivity for bacterial over human topoisomerases and retains efficacy against fluoroquinolone and carbapenem-resistant clinical isolates. Together, these structural, biochemical, and microbiological insights elucidate BWC0977's broad-spectrum antibacterial activity and reduced vulnerability to resistance, establishing it as a promising next-generation antibiotic to address the global threat of antimicrobial resistance.

DNA gyrase and topoisomerase IV are essential, homologous, highly conserved bacterial type II topoisomerases and are established targets of many successful antibiotics. These enzymes resolve topological challenges during DNA replication and transcription[1–3]. DNA gyrase ($GyrA_2GyrB_2$) introduces negative supercoils to alleviate torsional strain ahead of the replication fork and transcription complexes in the DNA, while the topoisomerase IV ($ParC_2ParE_2$) primarily resolves precatenanes and untangles the replicated DNA strands[2,4,5]. Understanding the mechanisms of these enzymes and their inhibitors has been important in developing effective antibiotics.

Fluoroquinolones are a class of broad-spectrum antibiotics that inhibit DNA gyrase and topoisomerase IV, albeit with varying potencies depending on the species of Gram-positive and Gram-negative bacteria[6]. Their inhibitory mechanism involves two fluoroquinolone molecules binding to two cleaved DNA sites within the enzyme–DNA complex. This forms a stable cleavage-complex that prevents the DNA strands from re-ligating, resulting in double-strand DNA breaks and ultimately causing bacterial cell death. Fluoroquinolone resistance is often caused by key mutations in the targets, at conserved acidic residues that anchor the water-metal ion bridge that are essential for compound binding[7–11]. Mutations are frequently observed at GyrA

[1]Bugworks Research India Pvt. Ltd., Center for Cellular & Molecular Platforms, National Center for Biological Sciences, Bangalore, India. [2]The University of Trans-Disciplinary Health Sciences and Technology (TDU), Bengaluru, Karnataka, India. [3]Department of Life Science and Technology, Institute of Science Tokyo, Yokohama, Japan. [4]Institute for Protein Research,, Osaka University, Suita, Osaka, Japan. ✉e-mail: murakami@life.isct.ac.jp; nainesh@bugworksresearch.com

S83 and D87 (or E87) in Gram-negative gyrase and ParC S80 and E84 (or D84) in Gram-positive topoisomerase IV, resulting in less effective fluoroquinolone binding[11–14]. This reduced binding affinity means that higher concentrations of fluoroquinolones are required to achieve the same inhibitory effect, suggesting that both bridge-anchoring residues play a similar role in gyrase's affinity for the drug in *Escherichia coli, Acinetobacter baumannii, Klebsiella pneumoniae, Neisseria gonorrhoeae, Staphylococcus aureus* and *Streptococcus pneumoniae*[12,15]. Despite the evolving resistance to existing drugs, the essential roles and conserved nature of DNA gyrase and topoisomerase IV make them promising targets for new antibiotics. NBTIs are being developed to interact with alternate, non-overlapping amino acids, offering a potential solution to overcome fluoroquinolone resistance, including target- and efflux-mediated mechanisms[16–20]. NXL101 was the first reported NBTI interacting with topoisomerase IV and gyrase differently than fluoroquinolones. This breakthrough, along with the cocrystal structure of another NBTI, GSK299423 with *S. aureus* gyrase[21], significantly advanced NBTI research. Along with these advances, and efforts by pharmaceutical giants like GSK, Actelion, RedEx, Novartis, AstraZeneca, and Roche to develop diverse chemical classes of topoisomerase inhibitors, only gepotidacin (triazaacenaphthylene)[22–25] and zoliflodacin (spiropyrimidinetrione)[26–28] are in clinical development for treating infections caused by *N. gonorrhoeae* (Fig. 1). Gepotidacin is an NBTI that effectively inhibits both DNA gyrase and topoisomerase IV in *N. gonorrhoeae*, while zoliflodacin, a non-NBTI, targets primarily the gyrase by binding at the DNA cleavage sites like fluoroquinolones. Both demonstrate antibacterial efficacy against this pathogen, including strains resistant to other antibiotics, such as those with the GyrA-S91F and D95G fluoroquinolone resistance mutations. However, in vitro studies indicate that substitution mutations in GyrB—D429N, K450T, or S467N leads to zoliflodacin resistance in Neisseria species[26]. Gepotidacin and zoliflodacin are effective against specific bacterial infections, but their broad-spectrum activity is limited[25,29–32]. Recently, gepotidacin received FDA approval for the treatment of uncomplicated urinary tract infections caused by susceptible isolates of *E. coli, K. pneumoniae, C. freundii, S. saprophyticus*, and *E. faecalis*[33]. Efforts to expand the NBTIs activity in general against Gram-negative bacteria have been met with other challenges like hERG-mediated cardiovascular toxicity, poor solubility, and unfavorable pharmacokinetic properties[16,17,34].

Our recent discovery of BWC0977 (Fig. 1), a pyrazino-oxazinone-based NBTI, represents a major advancement in antibiotic development to date[35]. BWC0977 demonstrated a potent minimum inhibitory concentration (MIC$_{90}$) range of 0.03–2 µg/mL against a diverse global panel of multidrug-resistant (MDR) Gram-negative bacteria, including Enterobacterales and non-fermenters, Gram-positive bacteria, anaerobes, and biothreat pathogens. Notably, this panel encompasses key pathogens from the World Health Organization (WHO) list of "global priority" threats, such as carbapenem-resistant *Enterobacteriaceae, Pseudomonas aeruginosa*, and *A. baumannii*, as well as methicillin-resistant *S. aureus* (MRSA). The carbapenem-resistant Enterobacterales, that includes *K. pneumoniae*, are of critical priority in the Bacterial Priority Pathogens List (BPPL) and pose an enormous threat to patients health worldwide (WHO BPPL 2024)[36,37]. The emergence of *K. pneumoniae* hypervirulence and resistance to last-line antibiotics, such as carbapenems, in hospital-acquired pneumonia and bloodstream infections, is responsible for numerous deaths[37,38].

BWC0977 maintains antibacterial efficacy against the clinical isolates resistant to fluoroquinolones, carbapenems, and colistin, highlighting its potential as a valuable antimicrobial agent in the fight against resistant infections[35]. NBTIs can achieve balanced inhibition of both gyrase and topoisomerase IV across species, enhancing their efficacy against Gram-negative and Gram-positive bacteria[39]. While structural analyses confirm NBTI interactions with DNA gyrases, the interactions with topoisomerase IV remain speculative[39–41]. This study reports the first structural analysis of *K. pneumoniae* topoisomerase IV complexed with BWC0977 at 3.05 Å resolution, alongside biochemical characterization revealing its potent and balanced dual-targeting mechanism. The features that underpin BWC0977's exceptional antibacterial potency and spectrum, with reduced susceptibility to clinical resistance, are also identified.

## Results
### Potent dual inhibition of gyrase and topoisomerase IV by BWC0977, primarily inducing single strand DNA breaks
DNA supercoiling assays for gyrase activity using relaxed plasmid DNA as a substrate, along with topoisomerase IV decatenation assays using kinetoplast DNA as a substrate, have demonstrated that BWC0977 exhibits exceptional potency. The IC$_{50}$ values were determined to be 0.004 µM for gyrase and 0.013 µM for topoisomerase IV, significantly outperforming both gepotidacin and ciprofloxacin (Table 1 and Fig. 2A). While gepotidacin shows balanced inhibition of gyrase and topoisomerase IV similar to BWC0977, it does so with higher IC$_{50}$ values of 0.77 µM for *E. coli* gyrase and 0.78 µM for *E. coli* topoisomerase IV. Ciprofloxacin, in contrast, shows a preference for *E. coli* gyrase with an IC$_{50}$ of 0.26 µM but exhibits weak inhibition of 16 µM for *E. coli* topoisomerase IV activity. BWC0977 demonstrates remarkable superiority over other NBTIs and fluoroquinolones in inhibiting the catalytic activities of gyrase and topoisomerase IV.

**Fig. 1 | Chemical structure of clinical NBTI compounds (BWC0977 and gepotidacin), ciprofloxacin and zoliflodacin.** BWC0977 is composed of a DNA binding moiety, 7-Fluoro-1-methylquinolin-2(1H)-one on the LHS, a central four atom linker and an enzyme binding moiety pyrazino-oxazinone linked to oxazolidinone on the RHS. Gepotidacin is composed of a triazaacenaphthylene on the LHS, a central basic nitrogen linker region and a pyranopyridine on the RHS. Zoliflodacin is composed of GyrB interacting pyrimidinetrione moiety and a methyl-oxazolidine-2-one group attached to benzisoxazole core. Ciprofloxacin is composed of a ketone group that interacts with DNA, a quinolone core and fluorine atom that interacts with gyrase.

**Table 1 | BWC0977 is a highly potent and selective inhibitor of bacterial type II topoisomerases**

| Compound | IC$_{50}$ in µM | | | |
|---|---|---|---|---|
| | E. coli gyrase supercoiling | E. coli topo IV decatenation | Human topo IIα decatenation | Human topo IIß decatenation |
| BWC0977 | 0.004 ± 0.001 | 0.013 ± 0.004 | 113 ± 13 | >200 |
| Gepotidacin | 0.77 ± 0.2 | 0.78 ± 0.3 | >200 | >200 |
| Compound 18c | 0.1[34] | 0.16[34] | ND | ND |
| Ciprofloxacin | 0.26 ± 0.02 | 16 ± 3.1 | >200 | >200 |
| Levofloxacin | 0.39 | 13 | >200 | ND |
| Moxifloxacin | 0.29 | 9.4 | >200 | ND |
| Zoliflodacin | 0.8 | 27 ± 12 | >200 | ND |
| Teniposide | ND | ND | 18 ± 5 | 92 |

The compound IC$_{50}$ values reported for BWC0977, ciprofloxacin and gepotidacin in the table are an average (±standard deviation) of three independent experiments. Compound 18c data is reported from the cited reference. Ciprofloxacin was used as positive reference compound in gyrase and topoisomerase IV assays. Teniposide was used as a positive reference compound in human topoisomerase II decatenation assays.
*ND* not determined.

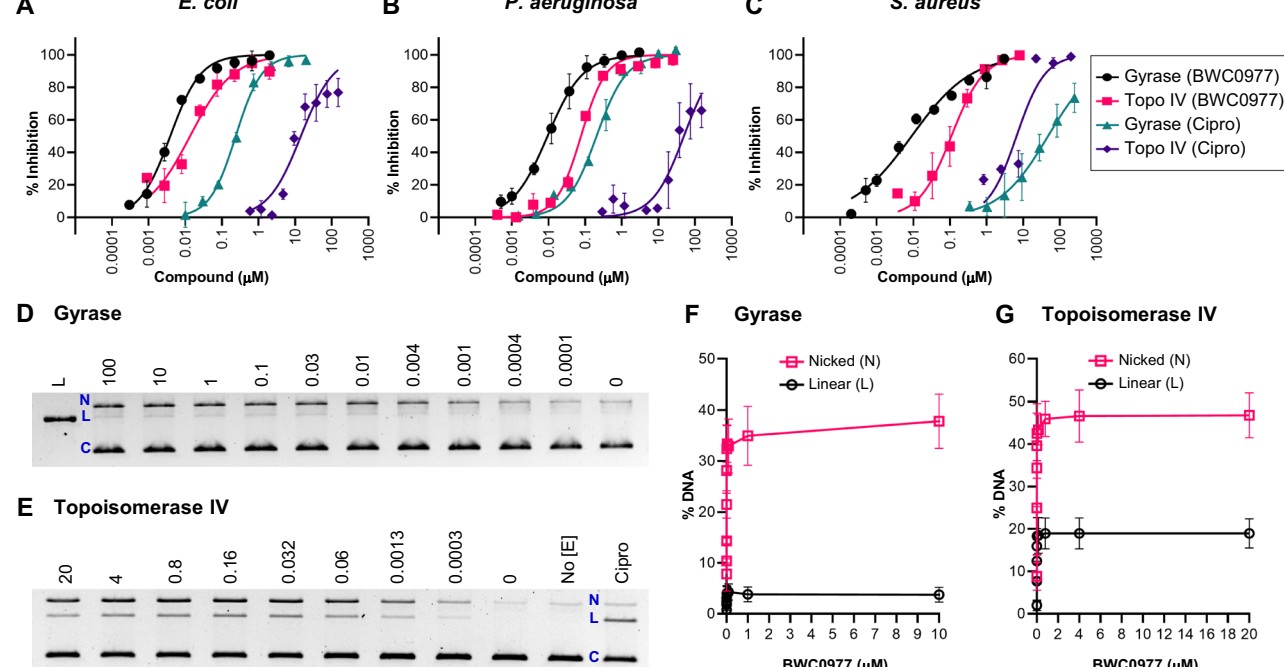

**Fig. 2 | Potent inhibition by BWC0977 on gyrase supercoiling and topoisomerase IV decatenation activities.** The graphs illustrate the inhibition potency of BWC0977 compared to ciprofloxacin on the DNA supercoiling activity of gyrases and the decatenation activity of topoisomerase IV from **A** *E. coli*, **B** *P. aeruginosa* and **C** *S. aureus*. The IC$_{50}$ values were calculated using appropriate reaction controls; maximum reaction (no compound, 100% enzyme activity) and minimum reaction (no enzyme, 0% enzyme activity). The error bars in the plots represent the standard deviation of three *E. coli* and *S. aureus* or two *P. aeruginosa* independent experiments. BWC0977 induces single strand breaks mediated by gyrase and topoisomerase IV. The gel images show BWC0977 concentration response (µM) performed with **D** *E. coli* gyrase and **E** *E. coli* topoisomerase IV in a DNA cleavage assay setup. The gel bands (L = linear DNA, N = nicked DNA, C = circular DNA) were quantified and plotted on GraphPad Prism as % DNA breaks induced by BWC0977 with **F** gyrase and **G** topoisomerase IV enzymes. The error bars represent the standard deviation of three independent experiments.

The nanomolar potency of BWC0977 extends to type II topoisomerases of *P. aeruginosa* and *S. aureus*, where ciprofloxacin shows limited activity, particularly against Gram-positive *S. aureus* gyrase (Fig. 2B, C and Supplementary Fig. 1A–G). Moreover, BWC0977 displays exceptional specificity, demonstrating over 5000-fold selectivity for bacterial enzymes over human topoisomerases IIα and IIβ (Table 1). This underscores its potential as a highly selective and potent therapeutic agent.

The cleavage-complex assays reveal that BWC0977 has a distinct mechanism of inhibition compared to fluoroquinolones. BWC0977 primarily induces high levels of single-strand DNA breaks in a concentration-dependent manner (corresponding to nicked DNA in Fig. 2D–G) with both gyrase and topoisomerase IV, even at low nanomolar compound concentrations. In contrast, ciprofloxacin promotes double-strand DNA breaks (corresponding to linear DNA in Supplementary Fig. 2A–D)[42]. These findings highlight that BWC0977 differs from fluoroquinolones in its potencies on gyrase and topoisomerase IV catalytic activities, as well as their mechanisms of inhibition.

### X-ray crystal structure of *K. pneumoniae* topoisomerase IV-DNA-BWC0977 ternary complex reveals compound binding mode and interaction residues with ParC

NBTIs exhibit broad-spectrum antibacterial efficacy by targeting both gyrase and topoisomerase IV with high potency. While extensive studies have

elucidated gyrase structures, no cocrystal structures of topoisomerase IV from Gram-negative bacteria complexed with an NBTI have been reported. To bridge this gap and better understand the potent inhibitory activity of BWC0977, we conducted a cocrystal structure analysis of *K. pneumoniae* topoisomerase IV. The structural characterization of *K. pneumoniae* topoisomerase IV is particularly valuable because this clinically significant human pathogen is known for its resistance to current therapies. Building on prior structural studies, we engineered a single-polypeptide fusion protein (ParEC core) derived from *K. pneumoniae* topoisomerase IV[43]. This construct integrates the C-terminal domain of ParE (residues 400−631) that consists of GHKL, transducer and TOPRIM domain with the N-terminal domain of ParC (residues 1–488) that consists of WHD, TOWER and coiled-coil domain, linked via a Glutamate-Serine (ES) linker. The resulting ParE$_2$C$_2$-equivalent dimer was optimized for crystallographic analysis.

We obtained a ternary complex of the ParEC core dimer, a 26-bp DNA duplex, and BWC0977, which crystallized at 3.05 Å resolution (PDB: 9KGT, Fig. 3A, B, Data collection and structure refinement statistics are summarized in Table 2). This represents the highest-resolution structure of *K. pneumoniae* topoisomerase IV bound to an NBTI to date. The asymmetric unit contains two copies of the ParEC-core dimer, each bound to double-stranded DNA and BWC0977 (Fig. 3A, B). Structural alignment of these complexes showed a high degree of similarity (RMSD ~ 1 Å), with minor conformational differences observed between protomers within the ParEC-core dimer (Supplementary Table 1). Unbiased electron density maps (Supplementary Fig. 3) identified a single BWC0977 molecule intercalated

between DNA bases at positions +2 and +3 (between A and T base pairs), forming a stable topoisomerase IV–DNA–BWC0977 ternary complex (Figs. 3 and 4). This binding mode is different from that of ciprofloxacin, moxifloxacin (PDB: 2XKK) and zoliflodacin (PDB: 8BP2), which exhibit two molecules bound at the cleavage sites (Supplementary Fig. 4). The 7-Fluoro-1-methylquinolin-2(1H)-one ring (LHS) of BWC0977 intercalates between DNA bases, engaging in π–electron interactions (Fig. 4A, B and Supplementary Table 2). Notably, the bicyclic LHS (rings C and D) participates in strong stacking interactions with the DNA bases, A and T (Fig. 4). The right-hand side (RHS) of BWC0977 consists of pyrazino-oxazinone and oxazolidinone moieties, which interact with key residues D79, M118, and R119 of ParC (Fig. 3C, D and Supplementary Table 2). This interaction region is located along the twofold symmetry axis of the ParEC dimer near the N-terminal WHD of ParE and it plays a critical role in enzyme inhibition. The -NH group of the pyrazino-oxazinone moiety forms a hydrogen bond with the ParC D79 from one subunit, while the oxazolidinone nitrogen interacts with the catalytically important R119 and while maintaining a weak interaction with the D79 from the second ParC subunit. These interactions stabilize the closed DNA gate conformation, effectively inhibiting the topoisomerase IV activity. Further stabilization arises from π–sulfur interactions between M118 and the pyrazino-oxazinone moiety (Fig. 3D). The oxygen atoms of the oxazolidinone ring contribute additional hydrogen bonds to both the side and main chains of R119, while R119 and Y120 further anchor BWC0977. Notably, R119 is conserved across type II topoisomerases and is essential for cleavage-religation activity[44]. Unique to

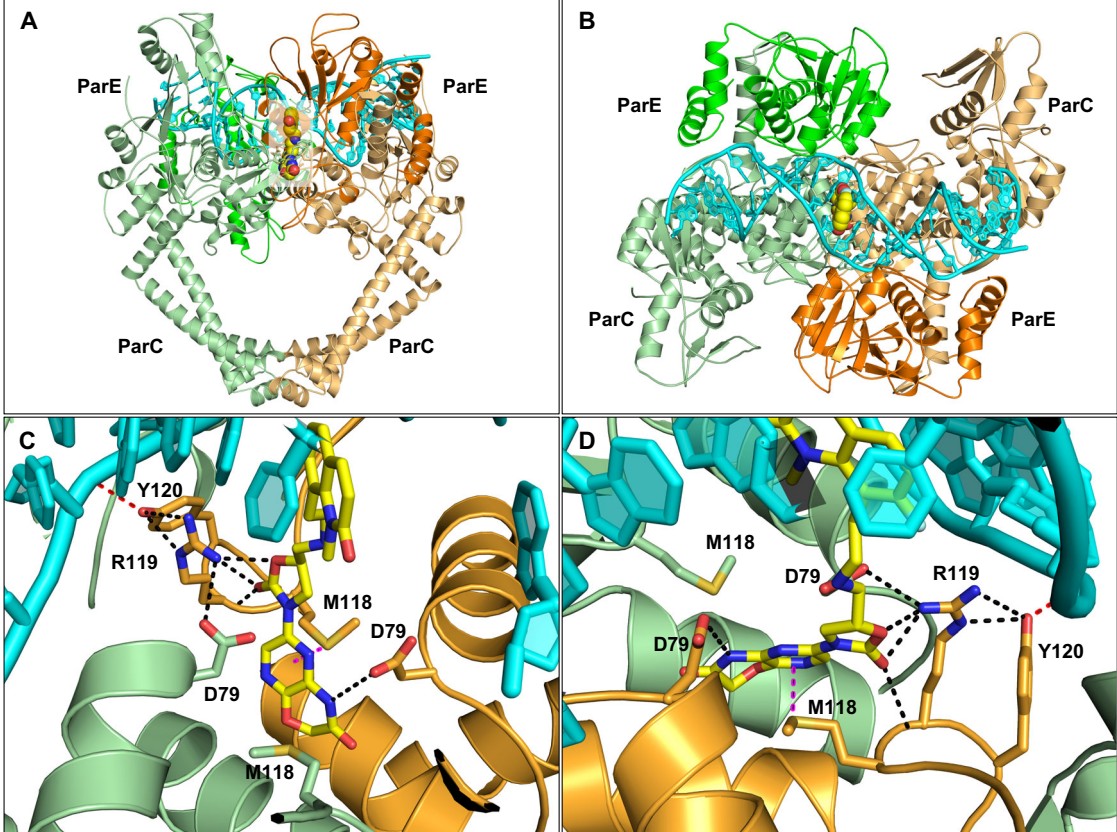

**Fig. 3 | *K. pneumoniae* topoisomerase IV in ternary complex with BWC0977 and DNA homoduplex at 3.05 Å resolution.** The crystal structure of BWC0977 bound to the ParEC core dimer and DNA homoduplex (PDB: 9KGT). **A, B** Orthogonal views of the ternary complex. Each monomer (ParEC core) in the functional dimer is represented in orange and green (ParE and ParC are in dark and pale shades, respectively) colors. BWC0977 is shown in CPK representation with the position of the C, N, O and F atoms in yellow, blue, red and light green, and DNA as a cartoon in cyan. The interaction of BWC0977, the pyrazino-oxazinone and oxazolidinone

moieties (RHS) at the ParEC core dimer interface. **C** The pyrazino-oxazinone and oxazolidinone moieties (RHS) of BWC0977 form many interactions with the binding site residues at the dimer interface, as shown here as dashed lines. Sulfur-aromatic interactions with both rings of the pyrazino-oxazinone (magenta dashed lines) and hydrogen bonds (black dashed lines) between the residues, including both subunits of the dimer. **D** The interaction between the catalytic residue Y120 and phosphate group (DNA) is also shown (red dashed lines) in the top view.

oxazolidinone-based compounds, this ring interacts with both the side and main chain NH of R119, and the guanidino group of R119, disrupting the DNA cleavage-religation cycle and inhibiting catalytic activity.

## Table 2 | Data processing and refinement statistics

| Data collection | *K. pneumoniae* ParEC-core (PDB ID: 9KGT) |
|---|---|
| Beam line | BL44XU, SPring-8, Japan |
| Space group | $P2_1$ |
| Cell dimensions *a, b, c* (Å) | 95.44, 163.27, 145.84 |
| $\alpha, \beta, \gamma$ (°) | 90, 94.55, 90 |
| Wavelength (Å) | 0.90000 |
| Resolution (Å) | 48.46 − 3.05 (3.10 − 3.05) |
| No. reflections | |
| Observed | 623,050 (29,734) |
| Unique | 84,963 (4199) |
| Redundancy | 7.33(7.08) |
| Rmerge | 0.096 (>1.0) |
| CC1/2 | 0.998 (0.574) |
| Completeness (%) | 99.90 (98.99) |
| I/σ (I) | 11.1 (1.0) |
| Refinement | |
| Resolution (Å) | 46.46 − 3.05 |
| $R/R_{free}$ | 0.223/0.270 |
| RMS deviation from ideal | |
| Bond lengths (Å) | 0.002 |
| Bond angles (°) | 0.550 |
| Average B factors (Å) | 149.9 |
| Ramachandran plot (%) | |
| Favored | 92.94 |
| Allowed | 6.96 |
| Disallowed | 0.10 |

Values in parentheses are for the highest resolution shell.

Quaternary structural comparisons revealed that BWC0977-bound complexes exhibit tighter packing about the twofold symmetry axis, with ParEC dimers compacted by ~2 Å relative to compound 34 (Quinolin-2(1H)-one scaffold)-bound structures[43] (PDB 6WAA, Supplementary Fig. 5A–D). For instance, the phosphate atom distance at the 16th DNA base decreased from 20.8 Å in compound 34 bound complexes to 16.9 Å in the BWC0977-bound complex. Similarly, the Cα atom distance for ParE residue V175 was reduced from 57.7 to 55.5 Å, suggesting that this compaction constrains the enzyme's conformational flexibility, thereby enhancing inhibition. Compared to previous NBTIs, BWC0977 exhibits superior binding at the active site through stronger and more extensive hydrogen bonding.

To examine similarities and differences in binding interactions, we performed structural alignments of the *K. pneumoniae* topoisomerase IV-BWC0977 structure (PDB: 9KGT) with *S. aureus* gyrase structures bound to NBTI compounds—gepotidacin (PDB: 6QTK) and compound 17a (Roche, PDB: 7FVT). Based on gyrase supercoiling and topoisomerase IV decatenation inhibition data, most NBTIs exhibit comparable potency against both the enzymes. However, BWC0977 exhibits significantly greater topoisomerase inhibition potency across all species, exceeding that of gepotidacin and compound 17a by more than tenfold (Table 1 and Supplementary Table 1). Structural comparisons between BWC0977 and gepotidacin reveal that the oxazolidinone ring of BWC0977 forms two to three additional hydrogen bonds with the R119 side chain guanidino group as well as the main chain NH (Fig. 5A). Residue R119 (*K. pneumoniae* topoisomerase IV) is known to interact with the catalytic residue Y120 during the DNA cleavage-religation reaction[44,45]. In contrast, the amine linker of gepotidacin (Fig. 5B) forms only a single hydrogen bond with the carboxyl side chain of D83 (D79 in *K. pneumoniae* topoisomerase IV), while its pyranopyridine ring (RHS) stacks between the two M121 residues (M118 in *K. pneumoniae* topoisomerase IV). Unlike BWC0977, gepotidacin does not directly interact with the catalytically important arginine residue. The RHS of BWC0977 accommodates both stacking and hydrogen bonding interactions: its pyrazino-oxazinone core stacks with the M118 side chains, while the oxazinone NH forms a hydrogen bond with the D79 carboxyl side chain (Fig. 5A). The oxazolidinone ring helps achieve stronger enzyme binding affinities for BWC0977. The LHS of both BWC0977 (Fig. 5A) and gepotidacin (Fig. 5B) stack between AT base pairs of the DNA duplex in the cocrystal structures, 9KGT and 6QTK, respectively. Furthermore, ~20-fold higher inhibition potency of BWC0977 against *E. coli* gyrase and topoisomerase IV compared to compound 17a suggests that enhanced DNA stacking affinity of its bicyclic aromatic ring

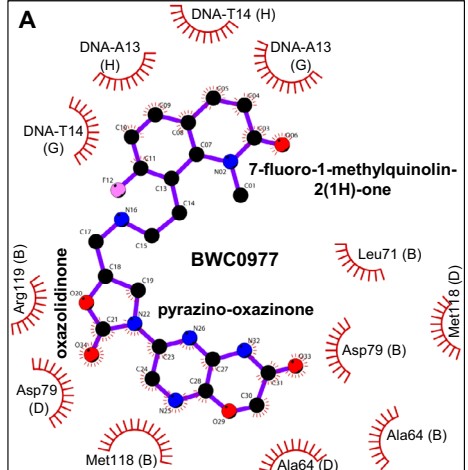

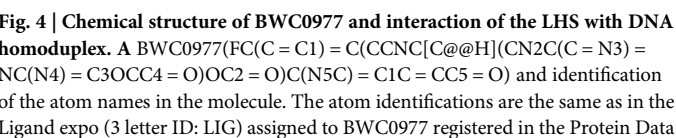

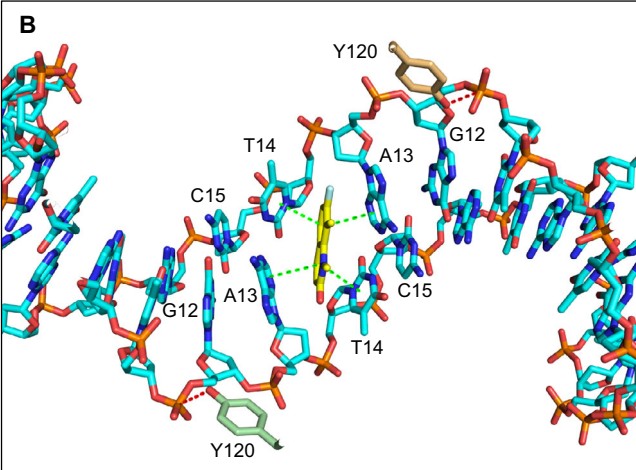

**Fig. 4 | Chemical structure of BWC0977 and interaction of the LHS with DNA homoduplex. A** BWC0977(FC(C = C1) = C(CCNC[C@@H](CN2C(C = N3) = NC(N4) = C3OCC4 = O)OC2 = O)C(N5C) = C1C = CC5 = O) and identification of the atom names in the molecule. The atom identifications are the same as in the Ligand expo (3 letter ID: LIG) assigned to BWC0977 registered in the Protein Data Bank (PDB: 9KGT). **B** Stabilization of LHS (7-Fluoro-1-methylquinolin-2(1H)-one) moiety by π–π stacking interaction with DNA base pairs (green dashed lines). The interaction between Y120 and phosphate, essential for enzyme activity, is also shown (red dashed lines).

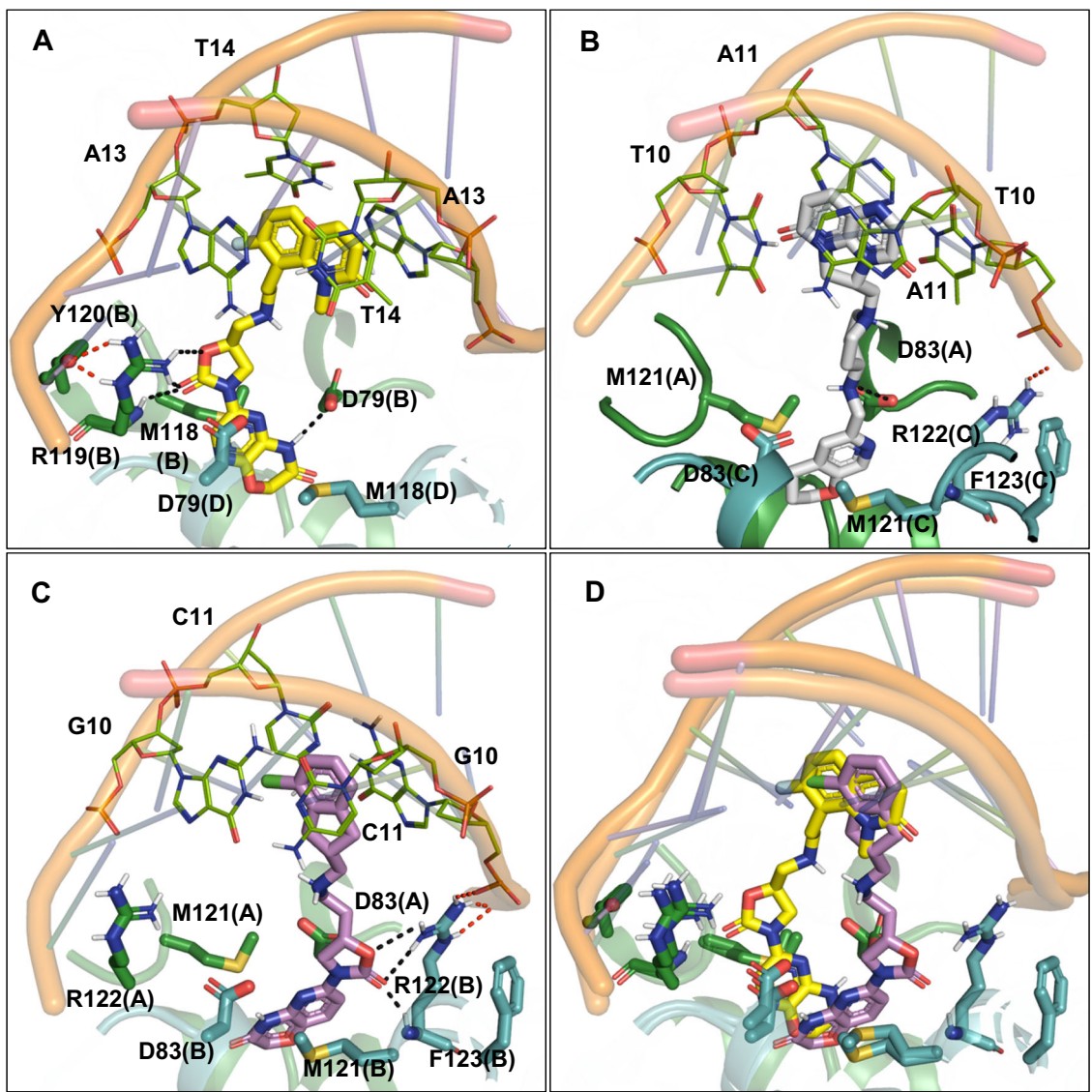

**Fig. 5 | Structural superposition of NBTI bound cocrystal structures. A** The BWC0977 cocrystal structure with *K. pneumoniae* topoisomerase IV (PDB: 9KGT) is used as a template for structural alignment with **B** gepotidacin (PDB: 6QTK) and **C** compound 17a, (Roche, PDB: 7FVT) cocrystals with *S. aureus* gyrase. **D** The structures of BWC0977 and compound 17a are superposed to compare binding mode similarities and differences. The H-bond interactions between the compound and binding site residues are highlighted in black dashed lines, while the intra-molecular interactions between the binding site residues and with DNA are shown in red dashed lines. Chain identifiers are highlighted in parentheses.

contributes to its superior activity. In contrast, compound 17a, which features a chloro-substituted single aromatic ring, engages in similar binding interactions but lacks the same level of stabilization (Fig. 5C, D).

### Reduced inhibition on mutant gyrase supercoiling activities confirms BWC0977's interaction residues

The inhibition of quinolone-resistant *E. coli* GyrA S83L, NBTI-resistant *E. coli* GyrA D82N (equivalent to *K. pneumoniae* ParC D79) and *E. coli* M120K (equivalent to *K. pneumoniae* ParC M118) mutant gyrases by BWC0977 and ciprofloxacin were measured in biochemical assays to confirm their interaction residues (Fig. 6). In supercoiling assays with these mutant gyrases, BWC0977 showed a >500-fold reduction in the inhibition potencies on GyrA D82N and M120K compared to wildtype (WT) gyrase. The mutations at these positions likely disrupt the interactions between BWC0977 and the enzyme, leading to a substantial decrease in inhibition. However, BWC0977 continued to be potent on quinolone-resistant S83L mutant gyrase activity (<3-fold relative to WT) and is consistent with the lack of cross-resistance observed in minimum inhibitory concentration

(MIC) experiments[35]. These results show that BWC0977 interacts with the gyrase D82 and M120, which are crucial for its inhibitory activity. Furthermore, our structural analysis (PDB: 9KGT) also confirms that BWC0977 interacts with topoisomerase IV D79 and M118 but not with S80 (Fig. 3C, D), which is in good agreement with the biochemical assay results.

### Balanced dual-targeting of gyrase and topoisomerase IV: mechanistic insights into BWC0977 resistance via target mutations in *E. coli* MG1655

We examined the interactions of BWC0977 in 9KGT and identified three important interacting amino acids in *E. coli* MG1655 topoisomerase IV (ParC D79, M118, R119) and gyrase (GyrA D82, M120, R121). Mutation studies were conducted to confirm these interactions. Substitutions at these sites could impact BWC0977 binding, lead to shifts in the MIC, and aid in the study of potential resistance mechanisms. Mutants of *gyrA* or *parC* alone, as well as strains with mutations in both genes to obtain amino acid substitutions in *E. coli* MG1655 (as listed in Table 3), were generated using an ssDNA recombineering technique[46].

**Fig. 6 | Inhibition on supercoiling activity of D82N, M120K mutant gyrases and ciprofloxacin resistant S83L gyrase by BWC0977 in comparison to wildtype (WT) gyrase. A** BWC0977 and **B** Ciprofloxacin concentration responses were performed with *E. coli* WT, GyrA D82N, GyrA S83L, and GyrA M120K mutant gyrases in DNA supercoiling assays using appropriate control reactions. The error bars in the IC$_{50}$ plots represent the standard deviation of three independent experiments.

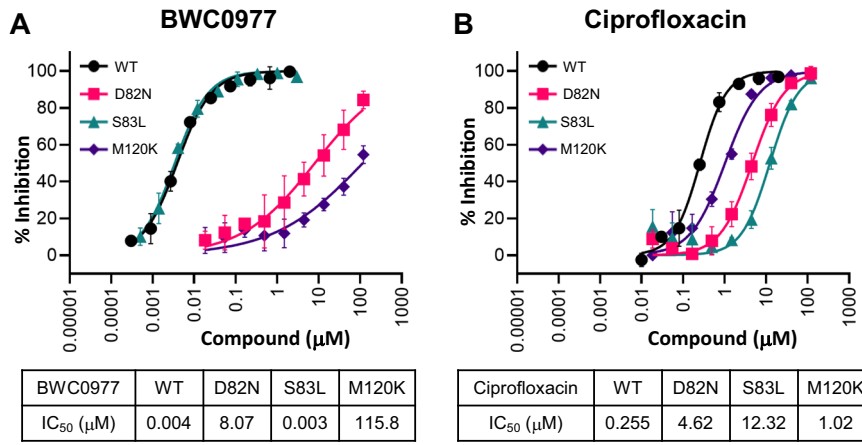

| BWC0977 | WT | D82N | S83L | M120K |
|---|---|---|---|---|
| IC$_{50}$ (µM) | 0.004 | 8.07 | 0.003 | 115.8 |

| Ciprofloxacin | WT | D82N | S83L | M120K |
|---|---|---|---|---|
| IC$_{50}$ (µM) | 0.255 | 4.62 | 12.32 | 1.02 |

## Table 3 | BWC0977's dual-target mechanism demonstrated using engineered *E. coli* MG1655 *gyrA* (gyrase) and *parC* (topoisomerase IV) mutant strains

| Substitution mutation | | MIC (µg/mL) | | | | |
|---|---|---|---|---|---|---|
| GyrA | ParC | BWC0977 | Gepotidacin | Ciprofloxacin | Levofloxacin | Ampicillin |
| WT | WT | 0.03 | 0.47 | 0.01 | 0.03 | 5 |
| D82N | WT | 0.08 | 0.42 | 0.08 | 0.13 | 5 |
| WT | D79N | 0.03 | 0.83 | 0.01 | 0.02 | 2.5 |
| D82N | D79N | 7.5 | >40 | 0.21 | 0.25 | 3.3 |
| M120A | M118A | 0.0625 | 0.63 | 0.01 | 0.02 | 4.2 |
| R121K | WT | 0.010 | 0.13 | 0.08 | 0.08 | 4.2 |
| WT | R119K | 0.026 | 0.42 | 0.021 | 0.04 | 5 |
| R121K | R119K | Synthetically lethal/nonviable | | | | |

The MIC of BWC0977 is shown with *E. coli gyrA* and *parC* mutant strains, and compared with gepotidacin, ciprofloxacin, and levofloxacin. Ampicillin is used as a control that does not target either gyrase or topoisomerase IV. The values reported in the table are an average of at least three independent experiments.

The strains with a single residue mutation at either GyrA D82N or ParC D79N showed only a marginal effect on MIC values, whereas the simultaneous mutations of both GyrA D82N and ParC D79N conferred a >200-fold reduced sensitivity to BWC0977, consistent with the structural data showing these residues are key for stabilizing the inhibitor-enzyme-DNA complex. Similarly, the double mutant strain GyrA M120A and ParC M118A was constructed; however, this alanine substitution did not affect the overall binding of BWC0977 (Table 3). While the *E. coli* M120K mutant gyrase enzyme exhibited a loss of inhibition by BWC0977 (Fig. 6A), the double mutant GyrA M120K and ParC M118K were nonviable (Supplementary Table 3).

Studies on the topoisomerase IV cleavage-religation mechanism, based on crystallography, emphasize the importance of arginine (*S. pneumoniae* R117) in assisting tyrosine (*S. pneumoniae* Y118) mediated catalytic activity[21,44,45]. This arginine is highly conserved and proposed to be critical for the topoisomerase activity. Consequently, we observed that GyrA R121K mutant exhibited reduced fitness relative to the wild-type and ParC R119K mutant strain in a MIC-based fitness study (Supplementary Fig. 6A–C). Although the arginine to lysine substitution in a single target was tolerated and did not affect BWC0977's cellular activity, the double mutant strain with GyrA R121K and ParC R119K could not be generated. Likewise, replacing arginine with alanine was not viable, indicating that substitutions at this position are poorly tolerated by the enzyme. The *S. aureus* truncated gyrase with R122A is reported to lack DNA cleavage activity[47].

### In vitro frequency of resistance studies helps understand BWC0977's low resistance frequency in Gram-negative bacteria

The in vitro spontaneous resistance frequencies of BWC0977 against various Gram-negative bacteria were determined, and resistant colonies were characterized using MIC determination and by sequencing of *gyrA*, *parC*, *gyrB*, and *parE* to identify mutations in the target binding sites. Previously, we reported that exposure of *E. coli* ATCC 25922 and *P. aeruginosa* ATCC 27853 to >4× MIC of BWC0977 yielded no resistant colonies. The in vitro resistance frequency studies of BWC0977 at 4× MIC against *E. coli* ATCC 25922, *P. aeruginosa* ATCC 27853, and *A. baumannii* ATCC 19606 indicated a resistance frequency of less than $1 \times 10^{-9}$, consistent with previous reports[35]. The sequencing of colonies obtained in these experiments revealed no mutations in key NBTI binding sites, suggesting that BWC0977 does not induce target site mutations. Whereas ciprofloxacin resistance frequency was $5 \times 10^{-9}$ with *E. coli* ATCC 25922 at 0.05 µg/mL (4× MIC), showing mutations in the key binding residues S83L or D87Y of GyrA subunit of the gyrase. However, these mutants remained sensitive to BWC0977.

Only *A. baumannii* ATCC 19606 exposed to 0.6 µg/mL (4× MIC) BWC0977 yielded five resistant mutant colonies. One of them exhibited a four-fold MIC increase, while the remaining four displayed shifts within twofold. None of these resistant mutants exhibited cross-resistance to other antibiotics (Supplementary Table 4). Sequencing of the *A. baumannii* BWC0977-resistant mutant revealed an M118R substitution in GyrA, with ParC, GyrB, and ParE remaining unchanged. The large side chain of the arginine in this mutant does not appear to cause a steric hindrance; it likely disturbs the M118 π–sulfur interactions that lead to a small shift in MIC, suggesting that this alteration may marginally reduce BWC0977 binding efficiency. In contrast, exposure of 2.4 µg/mL (4× MIC) ciprofloxacin resulted in the emergence of resistant colonies with MIC increases ranging from 4- to 32-fold and carried a G79C mutation in the GyrA subunit. These findings reinforce the distinct binding and inhibition mechanisms of BWC0977 compared with fluoroquinolones.

## Discussion

Antimicrobial resistance is a growing global health crisis, highlighting the urgent need for innovative broad-spectrum antibiotics that combat resistant bacteria and minimize the emergence of new resistant strains[48–50]. BWC0977, a novel bacterial topoisomerase inhibitor currently in clinical development, holds significant promise in this regard. Its dual-targeting mechanism, which inhibits both DNA gyrase and topoisomerase IV with low nanomolar potency, ensures strong antibacterial activity against a broad range of Gram-positive and Gram-negative bacteria. BWC0977 primarily induces single-strand DNA breaks mediated by these enzymes in contrast to fluoroquinolones. Importantly, it is highly selective for bacterial targets and does not inhibit human topoisomerases IIα and IIβ, thereby reducing the risk of target-mediated toxicity despite their structural similarities to bacterial type II topoisomerases[51–53].

NBTIs are known for their potent Gram-positive antibacterial activity, achieved by targeting DNA gyrase with higher potency than fluoroquinolones[39,40]. Consequently, their interactions were examined using *S. aureus* gyrase cocrystal structures. Gepotidacin (6QTK) structural studies show that it interacts with the GyrA D83 via direct and water-mediated hydrogen bonds[22,27]. While NBTIs are speculated to interact similarly with topoisomerase IV, structural evidence has been lacking, until now[6,41].

We report the 3.05 Å resolution cocrystal structure of BWC0977 bound to *K. pneumoniae* topoisomerase IV (PDB: 9KGT), captured in a ternary complex with a DNA duplex. The structure reveals a single BWC0977 molecule binding at the ParEC dimer interface, engaging key ParC residues—D79, M118, and R119. Unlike gepotidacin, which forms limited interactions, BWC0977's pyrazino-oxazinone and oxazolidinone moieties establish multiple stable contacts within the ParC binding pocket. A four-atom linker between the LHS and the oxazolidinone-linked RHS facilitates optimal molecular orientation, enabling simultaneous interactions with both DNA bases and critical ParC residues. Notably, while fluoroquinolone binding relies on metal-ion-mediated interactions with acidic residues, BWC0977 bypasses this requirement entirely, engaging topoisomerase IV through non–metal-ion-dependent mechanisms. The 9KGT structure provides a view of the oxazolidinone ring directly interacting with R119—an essential residue involved in the DNA cleavage-religation cycle. This ring engages in multivalent interactions with the guanidino side chain of R119, which in turn interacts with the hydroxyl group of Y120 that mediates nucleophilic attack on the DNA phosphate backbone[45]. Such a binding pattern is distinctive to oxazolidinone-based

inhibitors like BWC0977 and is not achievable with aminopiperidine-based compounds such as gepotidacin (Fig. 5A, B). The 9KGT structure advances our understanding of NBTI-mediated catalytic inhibition of Gram-negative topoisomerase IV and provides insights to develop new antibiotics to combat antimicrobial resistance.

The conservation of BWC0977 interaction residues across, and even beyond, ESKAPEE pathogens (Supplementary Fig. 7) suggests that it may strongly inhibit *K. pneumoniae* gyrase and topoisomerase IV enzymes. Its interactions with highly conserved, catalytically essential residues likely contribute to the potent, broad-spectrum antibacterial activity observed. These findings align with the superior antibacterial potency (MIC$_{90}$) of BWC0977 compared to gepotidacin and ciprofloxacin across a globally diverse panel of clinical Gram-positive and Gram-negative isolates—including both drug-sensitive and multidrug-resistant strains[35] (Table 4 and Supplementary Table 5).

The findings with the mutant enzymes indicate that mutations at individual GyrA interaction residues (D82N or M120K) completely abolished BWC0977's ability to inhibit gyrase's supercoiling activities. However, MIC data from single (GyrA D82N or ParC D79N) and double-target mutants (GyrA D82N + ParC D79N) suggest that a significant reduction in susceptibility to BWC0977 requires simultaneous mutations in both gyrase and topoisomerase IV. This likely explains the inability to obtain target mutants in *E. coli*, *P. aeruginosa*, and *A. baumannii*, further highlighting the balanced dual-targeting mechanism of BWC0977. Developing resistance would require simultaneous mutations at the interaction sites in two essential enzymes, the probability of such an event is low, given BWC0977's dual-target engagement at nanomolar potencies. This underscores BWC0977's robustness against resistance, equipping it with the potential to maintain clinical efficacy for a significant duration before resistance emerges.

## Methods

### DNA gyrase supercoiling assays with wildtype and mutant gyrase enzymes

The DNA supercoiling assays were performed using 1 nM *E. coli* wild-type gyrase holoenzyme and 60 ng relaxed pBR322 in a 30 μL reaction volume. The reaction buffer contained 35 mM Tris-HCl (pH 7.5), 24 mM KCl, 4 mM MgCl$_2$, 1 mM ATP, 2 mM DTT, 1.8 mM spermidine, 6.5% (w/v) glycerol, and 0.1 mg/mL bovine serum albumin. Assays using *E. coli* mutant gyrase enzymes included 6.7 nM GyrA S83L, 6 nM GyrA D82N, or 14.5 nM

## Table 4 | BWC0977 is superior to gepotidacin against sensitive and multidrug-resistant (MDR) bacterial strains

| Bacterial strains | MIC (μg/ml) | | |
|---|---|---|---|
| | BWC0977 | Gepotidacin | Ciprofloxacin |
| *K. pneumoniae* ATCC 13883 | 0.06 | 1 | 0.06 |
| *S. aureus* ATCC 29213 | 0.007 | 0.125 | 1 |
| *E. coli* ATCC 25922 | 0.060 | 0.5 | 0.007 |
| *A. baumannii* ATCC 19606 | 0.125 | 2 | 0.5 |
| *E. faecalis* ATCC 29212 | 0.03 | 1 | 0.5 |
| *P. aeruginosa* ATCC 27853 | 0.125 | 1 | 0.06 |
| *E. coli* BAA 2471 | 0.5 | 2 | >4 |
| *E. coli* BAA 2469 | 0.01 | 0.5 | >4 |
| *K. pneumoniae* BAA 2782 | 0.125 | 1 | >4 |
| *K. pneumoniae* BAA 2784 | 0.125 | 1 | >4 |
| *A. baumannii* BAA 2885 | 0.5 | 2 | >4 |
| *K. pneumoniae* KPNIH1 | 1 | 8 | >4 |
| *P. aeruginosa* BAA 2794 | 0.125 | 0.5 | >4 |
| *P. aeruginosa* BAA 2797 | 0.25 | 1 | >4 |

The minimum inhibitory concentrations of BWC0977 compared with gepotidacin and ciprofloxacin against drug-sensitive and multidrug-resistant (MDR) strains from ATCC. Values represent the mean of two independent experiments. *E. coli* BAA 2471, *E. coli* BAA 2469, *K. pneumoniae* BAA 2782, *K. pneumoniae* KPNIH1, *P. aeruginosa* BAA 2794, and *P. aeruginosa* BAA 2797 are resistant to ciprofloxacin and meropenem. *K. pneumoniae* BAA 2784 and *A. baumannii* BAA 2885 are resistant to ciprofloxacin, meropenem, and colistin.

GyrA M120K, while *P. aeruginosa* gyrase was tested at 1.3 nM under the same buffer conditions.

Reactions were conducted at 37 °C for 40 min in the presence of a series of compound dilutions in DMSO with appropriate assay controls. To stop the reactions, 3.7 μL of a stop solution (a mix of 2% sodium dodecyl sulfate (3 μL) and 15 mg/mL Proteinase K (0.7 μL)) was added. Samples were then mixed with 4 μL of STEB loading buffer [40% (w/v) sucrose, 100 mM Tris-HCl (pH 8), 1 mM EDTA and 0.5 mg/mL bromophenol blue] and loaded onto 0.8% agarose gels prepared in 1× TBE buffer (100 mM Tris-borate, pH 8.3, and 2 mM EDTA). Gels were run slowly (1.75 V/cm) to separate different DNA bands and stained with 0.8 μg/mL ethidium bromide for 10 min, followed by destaining in milliQ water. The DNA bands were visualized and imaged using the Thermo iBright digital imaging system.

Supercoiling assays for *S. aureus* gyrase were performed with 4.9 nM holoenzyme and 75 ng relaxed pBR322 in a 30 μL reaction volume. The reaction buffer consisted of 40 mM HEPES-KOH (pH 7.6), 10 mM magnesium acetate, 10 mM DTT, 2 mM ATP, 500 mM potassium glutamate, and 0.05 mg/mL albumin. Reactions were incubated at 37 °C for 35 min in the presence of compounds. 10 μL of STEB and 20 μL of chloroform:isoamyl alcohol (24:1) were added to stop the reaction. The assay plate was vortexed and briefly centrifuged to separate the phases. The upper aqueous phase (28 μL) was loaded onto 0.8% agarose gels, which were run at 1.75 V in 1X TBE buffer. Gels were stained and imaged, and the DNA bands were quantified using Quantity One Basic software. IC$_{50}$ values, representing the compound concentration that inhibited supercoiling activity by 50%, were determined using a log[inhibitor] vs. response (four-parameter) nonlinear least-squares fit in GraphPad Prism.

### Topoisomerase IV decatenation assays
Topoisomerase IV decatenation reactions were performed using 2.5 nM *E. coli* enzyme and 60 ng kinetoplast DNA (kDNA) in an assay buffer containing 50 mM HEPES-KOH (pH 7.6), 100 mM potassium glutamate, 10 mM magnesium acetate, 10 mM DTT, 1 mM ATP, and 50 μg/mL albumin. Assays with *P. aeruginosa* topoisomerase IV (3 nM) were conducted in a similar buffer but with 50 mM HEPES-KOH (pH 7.9), 100 mM potassium glutamate, 6 mM magnesium acetate, 6 mM DTT, 1 mM ATP, 2 mM spermidine, and 50 μg/mL albumin. For *S. aureus* topoisomerase IV assay, the reactions contained 4 nM enzyme in an assay buffer composed of 50 mM Tris-HCl (pH 7.5), 5 mM MgCl$_2$, 5 mM DTT, 1.5 mM ATP, 350 mM potassium glutamate, and 0.05 mg/mL albumin. The procedures followed were consistent with those used in the supercoiling assays. Decatenated minicircles were resolved on 0.8% agarose gels, while the kDNA substrate remained in the wells. Gels were stained with 0.8 μg/mL ethidium bromide solution for 10 min, followed by destaining. DNA bands were visualized using a gel documentation system, and compound inhibition analysis was carried out using the same procedures described for the supercoiling assays.

### Human topoisomerase II decatenation assays
Human topo IIα (0.9 nM) and topo IIß (2.2 nM) decatenation assays were performed following the same protocol used for bacterial topoisomerase IV decatenation assays. The assay buffer was made of 50 mM Tris. HCl (pH 7.5), 125 mM NaCl, 10 mM MgCl$_2$, 5 mM DTT, and 100 μg/mL albumin.

### DNA gyrase and topoisomerase IV cleavage-complex assays
The enzymes were titrated with 90 ng supercoiled pBR322 in the presence of 10 μM ciprofloxacin and incubated at 37 °C for 60 min. The enzyme concentration required to generate 50–70% cleaved complexes (double-strand DNA breaks) with ciprofloxacin was determined and used in the subsequent mechanism of inhibition studies. For cleavage assays, a series of BWC0977 or ciprofloxacin dilutions in DMSO was incubated with 25 nM gyrase and 90 ng supercoiled pBR322 for 60 min at 37 °C in an ATP-free assay buffer containing 35 mM Tris-HCl (pH 7.5), 24 mM KCl, 4 mM MgCl$_2$, 2 mM DTT, 1.8 mM spermidine, 6.5% (w/v) glycerol, and 0.1 mg/mL albumin.

Cleavage assays with *E. coli* topoisomerase IV were conducted using 13 nM enzyme with 90 ng supercoiled pBR322 in an assay buffer composed of 40 mM HEPES-KOH (pH 7.6), 100 mM potassium glutamate, 10 mM magnesium acetate, 10 mM DTT, and 50 μg/mL albumin. Reactions were stopped by adding 3.8 μL of a stop solution containing 0.8 μL Proteinase K (20 mg/mL) and 3 μL SDS (2%), followed by incubation at 37 °C for 60 min. Afterward, 4 μL of STEB loading buffer was added, and samples were run on pre-stained 0.8% agarose gels containing 0.4 μg/mL ethidium bromide. Gels were imaged and the DNA bands (Nicked (N) = single-strand DNA breaks, Linear (L) = double-strand DNA breaks, C = circular DNA) were quantified using Quantity One software. The % nicked and linear DNA formed with varying compound concentrations was plotted for comparison.

All enzyme assay kits that included topoisomerases, substrate DNA, and assay buffers were obtained from Inspiralis, UK.

### Minimum inhibitory concentration (MIC) determination
The compound MICs against *E. coli* wildtype, *gyrA*, and *parC* mutant strains, various sensitive and MDR strains were determined by the broth microdilution method according to the Clinical and Laboratory Standards Institute guidelines. In brief, freshly grown bacterial cultures (3–7 × 10$^5$ CFU/ml) were added into a 96-well microtiter plate containing 3 μL of test compound dilutions and incubated at 37 °C for 16–18 h. The absorbance was measured at 600 nm, and the MICs were reported as the compound concentration that resulted in ≥80% growth inhibition.

### Cloning, protein expression and purification for X-ray crystallography
The *K. pneumoniae* fusion-truncated topoisomerase IV (ParE30-(Glu-Ser)-ParC55) used for crystallography was cloned into a pET-22b (Invitrogen) expression vector with a C-terminal 6-histidine tag for purification via immobilized metal affinity chromatography. The recombinant plasmid was transformed into *E. coli* BL21 (DE3), and the transformants were grown in LB medium at 25 °C. Protein expression was induced with 0.1 mM isopropyl-β-D-thiogalactopyranoside at an OD$_{610}$ of 0.6 for 8 h. The cells were harvested by centrifugation, resuspended in lysis buffer (20 mM Tris pH 8.0, 200 mM NaCl, 10% (v/v) glycerol, 1 mM Na-EDTA, 2 mM TCEP and a protease inhibitor cocktail) and lysed by sonication (Branson). Cell debris and membrane fractions were removed by sequential centrifugation at 27,000 × *g* for 10 min and 145,000 × *g* for 1 h. The clear lysate was supplemented with imidazole and MgCl$_2$ to respective final concentrations of 10 mM and 2 mM and incubated with Ni-Sepharose 6 (Cytiva) affinity resin at 4 °C for 1 h. The resin was washed with 50 mM imidazole wash buffer (20 mM Tris pH 8.0, 200 mM NaCl, 10% (v/v) glycerol, 2 mM TCEP, 50 mM imidazole), and the protein was eluted with 200 mM imidazole. The ParEC core fraction was collected and further purified by size-exclusion chromatography (Superdex-200 pg 16/60, Cytiva) using an ÄKTA Explorer 10S (Cytiva) in buffer (20 mM Tris-HCl pH 7.5, 100 mM NaCl, 2 mM TCEP). Peak fractions were pooled and concentrated using an Amicon stirred cell (Merck Millipore) with a 50 kDa molecular weight cut-off Omega ultrafiltration membrane disc filter (Pall Corporation) to a final protein concentration of 6 mg/mL for crystallization studies.

### DNA synthesis and purification
A 26-bp palindromic oligonucleotide (5'-TTACGTTGTATGATCATA-CAACGTAA-3') was synthesized by Fasmac. The freeze-dried DNA was dissolved in Tris-EDTA buffer (pH 8.0) and annealed to form double-stranded DNA (dsDNA). The dsDNA was then purified from single-stranded oligos using hydroxyapatite column chromatography (Bio-Rad) and desalted with HiTrap desalting chromatography in nuclease-free water.

### Crystallization of BWC0977 in complex with *K. pneumoniae* topoisomerase IV and DNA
The purified ParEC core protein (4 mg/mL) was mixed with dsDNA and BWC0977 in a molar ratio of 23 μM ParEC core: 28 μM dsDNA: 200 μM BCW0977 and incubated at 20 °C for 8 h. Ternary complex crystals were grown using the sitting-drop vapor diffusion technique at 20 °C. The

complex solution was mixed in a 2:1 ratio with a reservoir solution containing 17% polyethylene glycol 3350, 0.1 M sodium formate, and 100 mM Bis-Tris propane buffer (pH 8.5). Crystals grew within 7–14 days, reaching an optimal size of $0.3 \times 0.3 \times 0.1$ mm$^3$. These crystals were put through a series of soaking steps to attain a glycerol concentration of 30% (v/v) for cryo-protection and were then picked up using nylon loops (Hampton Research) and flash-cooled in nitrogen gas from Rigaku cryostat.

### Data collection, structure determination and refinement

Diffraction data sets were collected at 100 K using an EIGER hybrid photon-counting pixel-array detector (Dectris) on the BL44XU beamline at SPring-8. Diffraction images were processed using the XDS package[54,55], while further refinement was carried out with programs from Phenix[54–57]. Data collection and structure refinement statistics are summarized in Table 2. Native data were collected at a wavelength of 0.9000 Å. The crystal structure was solved by molecular replacement in PHASER using a *K. pneumoniae* topoisomerase IV structure (PDB 6WAA) as the search model[43,58]. The model was built using COOT and refined with Phenix[56,57,59,60]. Ramachandran analysis showed 96.2% of residues in the favored region, with only 0.03% classified as outliers, and a MolProbity score of 1.92[61].

### In vitro resistance frequency determination and characterization of target mediated resistance to BWC0977

Spontaneous resistance frequencies to BWC0977 were evaluated in *E. coli* ATCC 25922, *P. aeruginosa* ATCC 27853, and *A. baumannii* ATCC 19606. Mid-logarithmic phase cultures (~$10^9$ CFU/mL) were plated onto Luria-Bertani agar containing BWC0977 at 4×, 8×, and 16× the respective MICs. Plates were incubated at 37 °C for 24–36 h, and resistance frequencies were estimated by counting colony-forming units (CFUs) on the drug-containing plates. MICs of the isolated colonies were determined, and the *gyrA*, *gyrB*, *parC*, and *parE* genes were sequenced to assess for target-site mutations. For comparison, ciprofloxacin resistance frequencies were assessed in parallel using the same experimental protocol.

### Statistics and reproducibility

All experiments were conducted with at least three independent biological replicates. Technical replicates were excluded from statistical analyses.

Data expressed as mean ± standard deviation. Half-maximal inhibitory concentrations (IC$_{50}$) were calculated using a four-parameter nonlinear regression model (log[inhibitor] vs. response) in GraphPad Prism.

All experiments were reproducible, and representative results are shown from studies repeated a minimum of three times.

### Reporting summary

Further information on research design is available in the Nature Portfolio Reporting Summary linked to this article.

### Data availability

The coordinates and structure factors for the topoisomerase IV from *K. pneumoniae* in complex with DNA and BWC0977 have been deposited in the Protein Data Bank with accession code PDB ID: 9KGT. The data processing and refinement statistics are shown in Table 2, uncropped gel images are included in supplementary information file and source data is provided as Supplementary Data.

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

## Acknowledgements

Satoshi Murakami, Ui Okada, and Eiki Yamashita acknowledge support by Japan Society for the Promotion of Science (JSPS) KAKENHI Grant Numbers JP21H02412 (S.M.), JP22K06099 (U.O.), and JP21K19337 (Eiki Yamashita). The crystallography research was partially supported by Platform Project for Supporting Drug Discovery and Life Science Research—Basis for Supporting Innovative Drug Discovery and Life Science Research (BINDS) from AMED (JP20am0101072) and Joint Research Committee of Institute for Protein

Research, Osaka University. Synchrotron radiation experiments were performed at BL44XU of SPring-8 (2023A6500, 2023A6700, 2023B6500, 2023B6700, 2024A6500, 2024A6700, 2024B6500, and 2024B6700). This work was performed in part under the Collaborative Research Program as the Visiting Fellow of the Institute for Protein Research, Osaka University (VFCR1). We thank our colleagues at Bugworks India Pvt. Ltd.—Bhavana S, Savitha Raveendran, and Akshaya Ravishankar for their contributions to the MIC experiments, resistant mutant generation, and sequencing. We also thank Vindhya R. Gowtham for her assistance during the initial molecular biology and protein purification efforts, and Vasan Sambandamurthy for his careful review of the manuscript and valuable feedback. Additionally, we acknowledge Akos Nyerges and Csaba Pál from the Synthetic and Systems Biology Unit, Biological Research Centre, Szeged, for generating the target mutant strains used in our MIC studies. We acknowledge the Centre for Cellular and Molecular Platforms (C-CAMP), Bengaluru, which hosts the Bugworks Research laboratory as an incubated facility. Research reported in this manuscript is in part supported by CARB-X. CARB-X's funding for this project is provided in part with federal funds from the U.S. Department of Health and Human Services (HHS); Administration for Strategic Preparedness and Response; Biomedical Advanced Research and Development Authority; under agreement number: 75A50122C00028, and by awards from Wellcome (WT224842) and Germany's Federal Ministry of Research, Technology and Space (BMFTR). The content of this press release is solely the responsibility of the authors and does not necessarily represent the official views of CARB-X or any of its funders. Research reported in this manuscript is in part supported by BIRAC, Dept. of Biotechnology, Govt. of India via BIPP grant BT/BIPP0803/30/14.

## Author contributions

Radha Nandishaiah designed, performed and analyzed data of biochemical experiments, and MIC studies with target mutant strains. Nainesh Katagihallimath supervised and interpreted the in vitro study results. Nainesh Katagihallimath, Satoshi Murakami, Shahul Hameed P and Santanu Datta conceptualized the BWC0977 crystallography project. Satoshi Murakami led X-crystallography studies. Ui Okada performed molecular biology experiments for crystallography. Maho Aoki and Satoshi Murakami performed sample preparation and crystallization. Eiki Yamashita performed X-ray diffraction experiments. Eiki Yamashita and Satoshi Murakami analyzed the X-ray data. Satoshi Murakami built, refined and analyzed the structure model. Sudipta Sarma and Anirudh P Shanbhag performed initial molecular biology and protein purification experiments. Sreevalli Sarma and Vasanthi Ramachandran determined in vitro resistance frequency and characterized of the isolated mutants. Shahul Hameed P led the medicinal chemistry efforts to design BWC0977, and Ranga Rao led the synthetic chemistry efforts. Nagakumar Bharatham and Suryanarayanan Venkatesan executed molecular modeling and structural comparison studies. Balasubramanian V managed the BWC0977 discovery project. Santanu Datta made intellectual contributions to BWC0977 in vitro studies and manuscript preparation. Radha Nandishaiah, Satoshi Murakami, Shahul Hameed P and Nainesh Katagihallimath wrote the paper. All authors approved the final version of the manuscript.

## Competing interests

The authors Radha Nandishaiah, Nainesh Katagihallimath, Sudipta Sarma, Anirudh P Shanbhag, Sreevalli Sharma, Nagakumar Bharatham, Suryanarayanan Venkatesan, Shahul Hameed P, Ranga Rao, Vasanthi Ramachandran, V. Balasubramanian, and Santanu Datta declare their competing interests as either current or former employees or consultants of Bugworks Research India Pvt. Ltd. and hold stock options in the company. Additionally, Shahul Hameed P., Nagakumar Bharatham, Nainesh Katagihallimath, Sreevalli Sharma, Radha Nandishaiah, Vasanthi Ramachandran, and V. Balasubramanian are inventors on the patent WO2018225097A1, which covers the compound BWC0977 disclosed in this manuscript. Anirudh P Shanbhag is currently employed with Novartis Healthcare Pvt. Ltd., Hyderabad 500081. Satoshi Murakami, Ui Okada, Maho Aoki, and Eiki Yamashita declare no competing interests.
