## [Transparent Peer Review file · Communications Biology]

Structural interactions of BWC0977 with *Klebsiella pneumoniae* topoisomerase IV and biochemical basis of its broad-spectrum activity

Corresponding Author: Dr Nainesh Katagihallimath

This manuscript has been previously submitted at another journal. This document only contains information relating to versions considered at Communications Biology.

Version 0:

Reviewer comments:

Reviewer #1

(Remarks to the Author)

This ms describes structural and biochemical analyses of the antibiotic BW0977, an NBTI compound that targets bacterial DNA gyrase and DNA topoisomerase IV. The authors show a 3.05-Å crystal structure of the compound bound to a protein construct based on *Klebsiella pneumoniae* topo IV. This structure is validated by mutant studies, and frequency of resistance studies that show that BW0977 yields low resistance frequencies. Overall the work suggests that BW0977 is a compound with promise from a clinical perspective.

Generally speaking this is a straightforward piece of work describing experiments that appear to be thorough and competently executed. I just have a few comments for the authors:

1. There is a tendency for the authors to use boastful language in the ms, which can be off-putting.
2. The authors claim that this is the highest resolution structure of a Gram-negative topoisomerase bound to an NBTI. This may be correct (I have only done a limited check), but this is a fast-moving field and new structures are emerging all the time.
3. Similarly they claim that this is the first co-crystal structure of a Gram-negative topo IV complex with an NBTI. Again, the authors need to thoroughly check the literature on this. There are modelled structures (e.g. Kokot et al. 2021) and these do perhaps deserve a mention.
4. Fig. 4F & G would be better as semi-log plots.

Reviewer #2

(Remarks to the Author)

Generally this paper seems to have been constructed to avoid detailing exactly how the crystal structure was determined and to what extent this depended on the structure of the related structure pdb code: 7FVT.

This needs to be clearly explained before the paper is acceptable for publication.

The authors must revise the paper to improve its level of transparency and clarity to make this clear to the general reader.

In particular, whilst most NBTIs show two equivalent binding modes (related by the twofold axis of the complex - on which compounds sit), 7FVT has been interpreted in terms of a single binding mode (note, however, that the difference density for 7FVT suggests some presence of the second, twofold related, conformation). The authors of this manuscript need to correct the manuscript to answer the following points:

(i) The 'Unbiased electron density maps' must be clearly shown (line 163).

(ii) How many binding modes did these unbiased electron density maps show? One binding mode or two?

Obviously in any one complex the compound can only have one binding mode - but often in crystals the electron density maps clearly show the two equivalent binding modes.

If the maps only show one binding mode there should be a reason. Can the authors explain?

Also it is not clear why they have not compared their structure of *Klebsiella pneumoniae* topoisomerase IV with the structure of *Acinetobacter baumannii* topoisomerase IV with moxifloxacin (pdb code: 2XKK). This should be done to complement the comparisons with *Klebsiella pneumoniae* topoisomerase IV, pdb code: 6WAA.

Specific points

1. Line 44 change from:

'these enzymes and their inhibitors has been crucial in developing effective antibiotics'

to:

'these enzymes and their inhibitors has been important in developing new effective antibiotics'

Reason: Most antibiotics do not target DNA gyrase and topoisomerase IV

2. Line 53. Include original reference for water-metal-ion bridge.

' are essential for compound binding7–10.'

Wohlkonig, Alexandre, et al. "Structural basis of quinolone inhibition of type IIA topoisomerases and target-mediated resistance." *Nature structural & molecular biology* 17.9 (2010): 1152-1153.

3. Line 69. 'gepotidacin (trizaacenaphthylene)20–23'

Include reference on FDA approval of gepotidacin (e.g. Mullard, Asher. "New antibiotic for urinary tract infections nabs FDA approval." *Nature reviews. Drug discovery*.)

4. Line 153. Change from:

'This construct integrates the TOPRIM domain of the ParE terminal (residues 400–631) with the WHD (Winged-Helix Domain) and TOWER domains of the ParC N-terminal (residues 1–488)'

to:

'This construct integrates the TOPRIM (and small Greek Key: residues 527-564) domains of the ParE C-terminus (residues 400–631) with the WHD (Winged-Helix Domain), TOWER and the forked cradle + dimer domains of the ParC N-terminus (residues 1–488) ..'

Reason - accuracy.

5. Line 163. Show in an extended figure the:

'Unbiased electron density maps'

Reason: 'It is not clear in from the current structure how much the structure of 7ftv was used in interpreting the electron density maps.' This must be made clear.

6. Line 166-167 check Extended Data Fig.:

Compare with *A. baumannii* topo IV structure with moxifloxacin (2xkk).

'This binding mode is different from that of ciprofloxacin which has two molecules bound at the cleavage sites (Extended Data Fig. 3).' + compare with zoliflodacin structure (pdb code: 8BP2)

6. Change 'along' to 'about'.

188 packing along the two-fold symmetry axis, with ParEC dimers compacted by ~2 Å relative to

7 . Line 354-356. Clarify sentence;

354 While limited MIC upregulation may occur

355 due to non-target mediated changes such as efflux or permeability, developing resistance would

356 require simultaneous mutations at interaction sites in two essential enzymes.

354 While limited MIC upregulation may occur

355 due to non-target mediated changes such as efflux or permeability, developing target mediated resistance would

356 require simultaneous mutations at interaction sites in two essential enzymes.

8. Please clarify description of binding of PD 0305970

195 stronger and more extensive hydrogen bonding. For example, PD 030597041 exhibited limited ..

Re: Reference 41 - Laponogov et al., 2010.

This paper says:

'The best crystals were used to collect the native PD 0305970 datasets with the best resolution of 3.1 Å'

and also says that:

'Crystals soaked using other drugs as well as the crystals from which PD 0305970 was back-soaked out, diffracted to a lower resolution and did not show a clear and interpretable drug envelope in the 2Fobs-Fcalc maps after refinement'

However, the 'Levofloxacin' structure reported in the same paper is at 2.9Å (pdb code: 3K9F) - from a crystal which was apparently soaked to displace the compound.

Could the authors re-refine the reported Levofloxacin structure to check it is not really a PD0305790 structure?

Has the compound had really been displaced?

If the compound was not displaced - could they say something like:

'Incidentally, when we tried re-refining the 2.9Å structure (pdb code: 3K9F) with PD 0305790 (as opposed to the reported Levofloxacin we saw that ...

or if the density really clearly shows levofloxacin they should say that 'The maps clearly showed ..'

The resolution is similar to that they are using in refinement, this will give the readers confidence.]

9. Line 207. Please change 'is known' to 'is believed' or include a reference for this statement:
207 pneumoniae topoisomerase IV) is known to interact with the catalytic residue Y120 during the
208 DNA cleavage-religation reaction.

10. Please change the word 'key' for the word 'three'
245 We examined the interactions of BWC0977 in 9KGT and identified key interacting amino acids in

Reason: Supplementary Figure 9 in Bax et al., 2010 gives more interacting residues with GSK299423. Figure 4 suggests an additional contact with compound.
and Extended Data Fig 6. does not detail how ' the key residues for BWC0977 binding' were selected.

Methods.

Line 470-471. Consider changing:

' STEB loading buffer [40% (w/v) sucrose, 100 mM Tris-HCl 471 (pH 8), 0.5 mg/mL bromophenol blue, and 1 mM EDTA]'
to :

'STEB loading buffer [40% (w/v) sucrose, 100 mM Tris-HCl 471 (pH 8), 1 mM EDTA and 0.5 mg/mL bromophenol blue]'

Reason STEB = sucrose, tris, EDTA, bromophenol blue.

Line 482. Correct English.

change

' The assay plate was vortexed followed by a brief centrifuged to separate the phases.'

to:

'The assay plate was vortexed then briefly centrifuged to separate the phases.'

Line 586: Include reference for search model

6WAA reference 39) as the search model.

39 Skepper, Colin K., et al. "Topoisomerase inhibitors addressing fluoroquinolone resistance in Gram-negative bacteria."
Journal of Medicinal Chemistry 63.14 (2020): 7773-7816.

Include GESAMT reference:

E.Krissinel (2012), Enhanced Fold Recognition using Efficient Short Fragment Clustering, J. Mol. Biochem., 1(2) 76-85.
and CCP4 reference:

Reviewer #3

(Remarks to the Author)

The manuscript reports structural and biochemical insights into the mechanism of BWC0977, a novel bacterial topoisomerase inhibitor (NBTI) in clinical development. The authors provide high-resolution cocrystal structures of BWC0977 bound to *Klebsiella pneumoniae* topoisomerase IV, alongside in vitro biochemical and resistance assays. The dual-target inhibition profile (gyrase and topoisomerase IV) is interesting and reveals a differentiated mechanism from fluoroquinolones.

Searching for new antibacterials and elucidating the mode of action of new antibiotics is due to increasing bacterial resistance urgently needed; therefore the scope of this paper is very interesting. However, there are some issues that need to be addressed prior to considering publication in Nature Communications Biology.

- Line 60 – please check the sentence “Despite the evolving resistance to these targets, the essential roles and conserved nature of DNA gyrase and topoisomerase IV make them promising candidates for new antibiotics.” Does resistance really evolve to the target or do the authors mean the drugs (fluoroquinolones).

- Can you please update the introduction, as gepotidacin has been approved in March for treating uncomplicated UTIs caused by *E. coli*.

- Line 95 – correct the citation

- Include standard deviation in the Table 1

- We suggest to perform also the multi step resistance studies, with serial sub-MIC passages to evaluate how fast the resistance can evolve over time (passages). Please compare the data with the positive controls (gepotidacin and ciprofloxacin).

- In the literature has been proven that the MICs for NBTIs are substantially more potent in knock-out strains lacking efflux pumps and bacterial strains with increased permeability. We suggest to perform the MICs also against mutant strains to prove that efflux pump or permeability is not the major issue with this compound.

- Please clarify if all the enzyme assays kits have been obtained from Inspiralis or only the DNA gyrase and topoisomerase IV cleavage-complex assays.

Version 1:

Reviewer comments:

Reviewer #1

(Remarks to the Author)

I can confirm that the authors have satisfactorily addressed my comments.

Reviewer #2

(Remarks to the Author)

Report on rebutal for:

'Structural and biochemical insights into the mechanism of BWC0977 - a clinical stage broad-spectrum novel bacterial topoisomerase inhibitor'.

The authors have successfully rebutted the reviewers comments. The compound is undoubtedly of major interest and this article deserves publication.

However, I would recommend, before the article is published one or two minor changes to improve its clarity. The literature is full of wrong structures, many of which are at limited resolution and the authors do not seem to fully understand which structures are incorrect.

Also, GSK, have created some confusion in the literature with the name NBTI.

BWC0977 is an NBTI, similar to gepotidacin which is also an NBTI.

However, zoliflodacin is not an NBTI and binds at the same site as fluoroquinolones (as the authors correctly state).

The paper would be improved by separating out these two novel classes of antibacterials (lines 68-81) targeting DNA Gyrase and topoisomerase IV.

The NBTI geoptidacin was approved by the FDA in March 2025.

Zoliflodacin, which is due to go to the FDA in December 2025, has very different safety barriers to overcome before approval. hERG safety is not, to the best of my knowledge, an issue with zoliflodacin.

Please also note a missing reference:

Blower, Tim R., Benjamin H. Williamson, Robert J. Kerns, and James M. Berger. "Crystal structure and stability of gyrase–fluoroquinolone cleaved complexes from Mycobacterium tuberculosis." *Proceedings of the National Academy of Sciences* 113, no. 7 (2016): 1706-1713.

And please change Line 54:

topoisomerase IV, resulting in less effective fluoroquinolone binding^{12,13}.

To both include the new reference and to acknowledge Wohlkonig et al., 2010.

So it should read something like:

topoisomerase IV, resulting in less effective fluoroquinolone binding¹¹⁻¹⁴.

where reference 14 = Blower et al., 2016 (see above).

If the authors care about accuracy perhaps they could also say:

'the first published structure'.

I understand GSK have unpublished higher resolution NBTI structures with a gram negative topo IV.

Reviewer #3

(Remarks to the Author)

The authors have thoroughly addressed all the concerns raised in the first round of review.

The updated introduction, corrections to citations and data presentation, as well as the additional experiments suggested, have substantially strengthened the manuscript.

I recommend acceptance of the manuscript in its present form.

We thank the reviewers for their thoughtful feedback. We have considered each comment and revised the manuscript accordingly. Below, we provide detailed responses and the rationale to all the points raised. We hope these responses and accepted revisions address the concerns (the reviewer comments are highlighted in grey, and the document has 11 pages).

Reviewers comments:

Reviewer #1 (Remarks to the Author):

This ms describes structural and biochemical analyses of the antibiotic BW0977, an NBTI compound that targets bacterial DNA gyrase and DNA topoisomerase IV. The authors show a 3.05-Å crystal structure of the compound bound to a protein construct based on *Klebsiella pneumoniae* topo IV. This structure is validated by mutant studies, and frequency of resistance studies that show that BW0977 yields low resistance frequencies. Overall the work suggests that BW0977 is a compound with promise from a clinical perspective.

Generally speaking this is a straightforward piece of work describing experiments that appear to be thorough and competently executed. I just have a few comments for the authors:
1. There is a tendency for the authors to use boastful language in the ms, which can be off-putting.

2. The authors claim that this is the highest resolution structure of a Gram-negative topoisomerase bound to an NBTI. This may be correct (I have only done a limited check), but this is a fast-moving field and new structures are emerging all the time.

3. Similarly they claim that this is the first co-crystal structure of a Gram-negative topo IV complex with an NBTI. Again, the authors need to thoroughly check the literature on this. There are modelled structures (e.g. Kokot et al. 2021) and these do perhaps deserve a mention.

Combined response to comments 1, 2 and 3:

We appreciate the reviewer's feedback and have revised the statements that appeared too assertive. Regarding the structural claims, we conducted a literature review to determine whether any experimental co-crystal structures of NBTIs bound to *Klebsiella pneumoniae* topoisomerase IV or any Gram-negative topoisomerase IV have been reported. Based on this review, no such structures have been published to date. Moreover, the publications highlight the lack of such a structure. For instance:

- Collins & Osheroff (2024) explicitly state that while NBTIs are assumed to bind topoisomerase IV similarly to gyrase, **“structures with this enzyme have not yet been reported”** (PMID: 38564341).
- Cumming et al. (2023) also note that **“no experimental structure of an NBTI bound to topo IV has been reported,”** and modelling studies are used to infer binding due to high homology with gyrase (PMID: 37465290).

Below is a table of reported cocrystal structures of compounds (non-NBTIs except for BWC0977) with topoisomerase IV enzyme.

Sl. No	Compound	Resolution	Strain	protein	PDB code
1	Levofloxacin	3.35 Å	Kpn and Spn	Topoisomerase IV (ParE CTD 390–631 and ParC NTD 1–490 fused)	5EIX
2	compound 34 (quinolin-2(1H)-one series)	3.2 Å	Kpn	ParE CTD (residues 390–631) and NTD ParC (residues 1–490) linked by Glu-Phe linker	6WAA
3	compound 25	3.3 Å	Kpn	topo IV	7LHZ
4	PD 0305970 (Quinazolidione)	3.04 Å	Spn	topo IV	4Z4Q
5	BWC0977	3.05 Å	Kpn	topo IV	9KGT

We acknowledge the existence of modelled structures, such as those referred by Kokot et al. (2021). This reference along with a second publication from the same group had already been included in the manuscript (see ref. 38).

To avoid overstatement, we have removed the phrase “marking the highest resolution structure” and rephrased the statement. We hope these revisions address the reviewer’s concerns and improve the clarity and tone of the manuscript.

4. Fig. 4F & G would be better as semi-log plots.

We believe the semi-log plots referenced likely correspond to Fig. 2F and G. We have generated and included the semi-log versions of **Fig. 2F and G** as below for your reference. While semi-log plots can offer enhanced visualization of DNA percentage changes, most often the linear plots are used in cleavage assays to represent the formation of nicked (single-strand breaks) and linear (double-strand breaks) DNA species. This format allows for direct interpretation of the relative proportions of DNA forms resulting from compound treatment. To maintain consistency with the cited references (ref#21 and 61) in illustrating the mode of inhibition (MOI) of bacterial topoisomerases, we have opted to retain the linear plots in the main figure. This approach aligns with precedent set in prior publications, including:

- Gibson et al., 2019 (Ref#21), which uses linear plots to depict cleavage products in *Staphylococcus aureus* gyrase assays.
- Oviatt et al., 2024 (Ref#61), which similarly employs linear representations for *E. coli* gyrase and topoisomerase IV cleavage assays.

We hope this clarifies our rationale; however, we will be happy to include the semi-log plots as supplementary figures if it is deemed essential.

Figure 2F-G: The gel bands (L=linear DNA, N=nicked DNA, C=circular DNA) were quantified and plotted on GraphPad Prism as % DNA breaks induced by BWC0977 with (F) gyrase and (G) topoisomerase IV enzymes.

Reviewer #2 (Remarks to the Author):

Generally this paper seems to have been constructed to avoid detailing exactly how the crystal structure was determined and to what extent this depended on the structure of the related structure pdb code: 7FVT. This needs to be clearly explained before the paper is acceptable for publication. The authors must revise the paper to improve its level of transparency and clarity to make this clear to the general reader. In particular, whilst most NBTIs show two equivalent binding modes (related by the twofold axis of the complex - on which compounds sit), 7FVT has been interpreted in terms of a single binding mode (note, however, that the difference density for 7FVT suggests some presence of the second, twofold related, conformation). The authors of this manuscript need to correct the manuscript to answer the following points:

Please allow us to clarify that the crystal structure of BWC0977 bound to *Klebsiella pneumoniae* topoisomerase IV (PDB ID: 9KGT) was not based on or influenced by PDB ID: 7FVT, which is a structure of **Gram-positive gyrase**- from *Staphylococcus aureus*. Our study focuses exclusively on **topoisomerase IV**, and 7FVT was neither referenced nor used in any stage of model building or refinement. Instead, we used PDB ID: 6WAA, a *K. pneumoniae* topoisomerase IV structure as the search model for molecular replacement, as described in the Methods section.

(i) The 'Unbiased electron density maps' must be clearly shown (line 163).

We have now included the unbiased electron density maps in Extended Data Fig 3 (Lines 164-165), showing the clear placement of BWC0977 within the binding pocket.

(ii) How many binding modes did these unbiased electron density maps show? One binding mode or two?

Obviously in any one complex the compound can only have one binding mode - but often in crystals the electron density maps clearly show the two equivalent binding modes.

If the maps only show one binding mode there should be a reason. Can the authors explain?

The electron density maps revealed only one binding mode for BWC0977. No fragments of weak or ambiguous density suggesting a second conformation were observed. This is consistent with our co-crystallization approach, where BWC0977 was added to the purified ParEC core prior to crystallization. A single molecule of asymmetric BWC0977 binds and induces asymmetry at the twofold symmetry axis of the ParEC dimer. Therefore, the twofold symmetry axis around the ParEC dimer in the enzyme-BWC0977 complex is no longer strictly a twofold symmetry axis.

Additionally, the linker region of BWC0977, comprising a positively charged amine and an oxazolidinone moiety, enables the molecule to adopt a curved binding mode, facilitating additional interactions with the enzyme; particularly with R119. These unique functionalities of the compound might contribute to the observed single binding orientation, bowing away from the twofold axis (Ref. Figure 3C & 3D and Figure 5A & 5D in the main manuscript).

Also it is not clear why they have not compared their structure of *Klebsiella pneumoniae* topoisomerase IV with the structure of *Acinetobacter baumannii* topoisomerase IV with

moxifloxacin (pdb code: 2XKK). This should be done to complement the comparisons with *Klebsiella pneumoniae* topoisomerase IV, pdb code: 6WAA.

We have compared the binding modes of BWC0977 (9KGT), ciprofloxacin (5BTC), compound 34 (6WAA), moxifloxacin (2XKK), and zoliflodacin (8BP2) in Extended Data Fig. 4A–D. Please note the correction in compound names (Extended Data Fig. 4A: NBTI is now Compound 7; Extended Data Fig. 4B: NBTI is now BWC0977, and ciprofloxacin is compound 34). We have compared BWC0977 (9KGT) with compound 34 (6WAA) extensively because they are both *K. pneumoniae* topoisomerase IV structures.

As per the reviewer's suggestion, we performed BWC0977 (9KGT with *K. pneumoniae* topo IV) and moxifloxacin (2XKK with *A. baumannii* topo IV) superimposition and is shown in Extended Data Fig. 4C. In the case of *Acinetobacter baumannii* topoisomerase IV complex with moxifloxacin (PDB code: 2XKK), two molecules of moxifloxacin bind at QRDR regions. The compound 34 and moxifloxacin (Extended Data Fig. 4B and C respectively) show binding in almost identical positions except for a few differences in interaction patterns at the QRDR region.

Extended Data Fig 4. Binding mode of ciprofloxacin, moxifloxacin, zoliflodacin and NBTIs shown with gyrase and topoisomerase IV models.

A single NBTI (compound 7, BWC0977) binds between the two cleavage sites at the dimer interface in both enzymes, whereas two molecules of ciprofloxacin, compound 34, moxifloxacin, or zoliflodacin bind at the DNA cleavage sites. In the models shown here, (A) *S. aureus* gyrase in complex with Compound 7 (PDB 5BS3) was superimposed on *Mycobacterium tuberculosis* gyrase in complex with ciprofloxacin (PDB 5BTC, not shown in the image). (B) *K. pneumoniae* topoisomerase IV in complex with BWC0977 (PDB 9KGT) was superimposed with *K. pneumoniae* topoisomerase IV in complex with compound 34 (PDB 6WAA, not shown in the image). (C) *K. pneumoniae* topoisomerase IV structure in complex with BWC0977 (PDB 9KGT) was superimposed with *A. baumannii* topoisomerase IV in complex with moxifloxacin (PDB 2XKK, not shown in the image). (D) *K. pneumoniae* topoisomerase IV-BWC0977 structure was superimposed on *S. aureus* gyrase in complex with zoliflodacin (PDB 8BP2, not shown in the image). Ciprofloxacin, compound 34, moxifloxacin and zoliflodacin molecules are shown as yellow spheres. The NBTIs, compound 7 and BWC0977 are shown as magenta spheres.

Specific points

1. Line 44 change from:

'these enzymes and their inhibitors has been crucial in developing effective antibiotics' to: 'these enzymes and their inhibitors has been important in developing new effective antibiotics' Reason: Most antibiotics do not target DNA gyrase and topoisomerase IV

Changed accordingly (Line 43).

2. Line 53. Include original reference for water-metal-ion bridge. 'are essential for compound binding' 7–10. 'Wohlkonig, Alexandre, et al. "Structural basis of quinolone inhibition of type IIA topoisomerases and target-mediated resistance." *Nature structural & molecular biology* 17.9 (2010): 1152-1153.

Original reference included as ref#11.

3. Line 69. 'gepotidacin (triazacenaphthylene)' 20–23' Include reference on FDA approval of gepotidacin (e.g. Mullard, Asher. "New antibiotic for urinary tract infections nabs FDA approval." *Nature reviews. Drug discovery.*)

The FDA approval is now mentioned in Lines 75-77 and included as ref#28.

4. Line 153. Change from:

'This construct integrates the TOPRIM domain of the ParE terminal (residues 400–631) with the WHD (Winged-Helix Domain) and TOWER domains of the ParC N-terminal (residues 1–488) '

to:

'This construct integrates the TOPRIM (and small Greek Key: residues 527-564) domains of the ParE C-terminus (residues 400–631) with the WHD (Winged-Helix Domain), TOWER and the forked cradle + dimer domains of the ParC N-terminus (residues 1–488)..'
Reason - accuracy.

Thank you for the correction and the following change is made in the lines 152-155 and in Figure 3A-B of the revised manuscript:

"This construct integrates the C-terminal domain of ParE (residues 400–631) that consists of GHKL, transducer and TOPRIM domain with the N-terminal domain of ParC (residues 1–488) that consists of WHD, TOWER and coiled coil domain, linked via a Glutamate-Serine (ES) linker."

5. Line 163. Show in an extended figure the: 'Unbiased electron density maps'

Reason: 'It is not clear in from the current structure how much the structure of 7fvt was used in interpreting the electron density maps.' This must be made clear.

Please note that **Extended Data Fig. 3** in the revised manuscript presents the unbiased electron density map used in our structural analysis. Importantly, we did not use the 7FVT structure for model building or interpretation of the electron density, as 7FVT represents *S. aureus* gyrase, whereas our study focuses on *K. pneumoniae* topoisomerase IV. Instead, we used the **6WAA** structure of *K. pneumoniae* topoisomerase IV as the search model for molecular replacement during structural analysis. The BWC0977 model was subsequently built independently from the omit map. The **polder map** shown in **Extended Data Fig. 3B**, is an omit map calculated using a protein model obtained by removing BWC0977 from the

refined model and the observed intensity data without phase (i.e., model bias free). The resulting map allowed us to confidently place BWC0977 at 3.05 Å resolution.

The 7FVT structure was used solely for comparative analysis in **Figure 5**, to examine and contrast the interaction pattern of BWC0977 with that of compound 17a bound to *S. aureus* gyrase (7FVT).

6. Line 166-167 check Extended Data Fig.: Compare with *A. baumannii* topo IV structure with moxifloxacin (2xkk). 'This binding mode is different from that of ciprofloxacin which has two molecules bound at the cleavage sites (Extended Data Fig. 3).' + compare with zoliflodacin structure (pdb code: 8BP2).

In the revised manuscript, Extended Data Fig. 3 has been updated and renamed Extended Data Fig. 4. This figure now includes panels illustrating the binding modes of various compounds.

Extended Data Fig. 4A shows the binding mode of the NBTI compound 7 (PDB ID: 5BS3) with *S. aureus* gyrase, compared to ciprofloxacin (PDB ID: 5BTC) with *M. tuberculosis* gyrase.

Extended Data Fig. 4B presents BWC0977 (PDB ID: 9KGT) compared to compound 34 (PDB ID: 6WAA) in the *K. pneumoniae* topoisomerase IV complex. Extended Data Fig. 4C depicts moxifloxacin with *A. baumannii* topoisomerase IV (PDB ID: 2XKK), and Extended Data Fig. 4D shows zoliflodacin with *S. aureus* gyrase (PDB ID: 8BP2), both compared to BWC0977 in the *K. pneumoniae* topoisomerase IV complex.

These structural comparisons highlight that NBTIs (BWC0977 and compound 7) bind at the dimer interface of gyrase or topoisomerase IV, while fluoroquinolones (ciprofloxacin, moxifloxacin) and compound 34 bind at the DNA cleavage sites. This reinforces our observation that two molecules of Quinolones bind to the Gyrase/TopoIV-DNA complex, whereas only one molecule of NBTI binds at the dimer interface.

In summary, all quinolone derivatives, including moxifloxacin and zoliflodacin (spiropyrimidinetrione), bind two molecules at the QRDR region, while NBTI compounds bind at the GyrAB or ParEC dimer interface. This clearly demonstrates the distinct binding patterns of Quinolones and NBTIs, with the former binding at the DNA cleavage sites and the latter at the dimer interface.

6a. Change 'along' to 'about'. 188 packing along the two-fold symmetry axis, with ParEC dimers compacted by ~2 Å relative to

'along' is changed to 'about' in line 190 of the revised manuscript

7. Line 354-356. Clarify sentence;

While limited MIC upregulation may occur due to non-target mediated changes such as efflux or permeability, developing resistance would require simultaneous mutations at interaction sites in two essential enzymes.

If an antibiotic targets two essential enzymes, the likelihood of both enzymes mutating simultaneously in the same cell is much lower. This is because the probabilities of the two independent mutations occurring are multiplied together, making the overall probability of

resistance developing significantly lower. If the independent probability of a single mutation is 10^{-7} , the probability of two independent mutations occurring simultaneously would be $10^{-7} \times 10^{-7} = 10^{-14}$. This greatly reduces the chances of resistance developing, which is why dual-targeting by BWC0977 could be more robust against resistance.

8. Please clarify description of binding of PD 0305970 stronger and more extensive hydrogen bonding. For example, PD 030597041 exhibited limited.. This paper says: 'The best crystals were used to collect the native PD 0305970 datasets with the best resolution of 3.1 Å' and also says that: 'Crystals soaked using other drugs as well as the crystals from which PD 0305970 was back-soaked out, diffracted to a lower resolution and did not show a clear and interpretable drug envelope in the 2Fobs-Fcalc maps after refinement' However, the 'Levofloxacin' structure reported in the same paper is at 2.9Å (pdb code: 3K9F) - from a crystal which was apparently soaked to displace the compound.

Could the authors re-refine the reported Levofloxacin structure to check it is not really a PD0305790 structure?

Has the compound had really been displaced?

If the compound was not displaced - could they say something like:

'Incidentally, when we tried re-refining the 2.9Å structure (pdb code: 3K9F) with PD 0305790 (as opposed to the reported Levofloxacin we saw that...

or if the density really clearly shows levofloxacin they should say that 'The maps clearly showed..' The resolution is similar to that they are using in refinement, this will give the readers confidence.]

We realize that our discussion on PD 0305970 caused some confusion. Our observation was that PD 0305970, a quinazoline-2,4-dione, does not form a productive interaction with the **conserved arginine** due to the relatively long distance. In contrast, BWC0977, an NBTI, shows a strong interaction between its oxazolidinone ring and ParC R119, mediated by multiple hydrogen bonds, suggesting a more robust and functionally relevant interaction. Since we mistakenly placed PD 0305970 in Line# 195 where BWC0977 was being compared with other NBTIs and caused confusion, we have removed the misleading lines of 195-197.

Further, we referred to reference #40 (now #43) and reference #41 (now #44) in Line 208 to highlight the role of the conserved arginine residue adjacent to the catalytic tyrosine, which assists in the DNA cleavage–religation reaction. Reference #43 documents the conservation of this arginine residue across multiple species, including GyrA R121 and ParC R119 in *E. coli*, R781 in *S. cerevisiae* topoisomerase II, and R804 and R825 in human topoisomerase II α and II β , respectively. Reference #44 elaborates on this catalytic role and reports the binding mode of PD 0305970, including its interaction with R117 in *S. pneumoniae* topoisomerase IV. Regarding the reviewer's comments on the resolution and refinement of the Levofloxacin structure (PDB code: 3K9F), we understand the concern. However, we did not perform soaking experiments. We used co-crystallization (of ParEC+DNA+BWC0977) to obtain the complex structure, ensuring no displacing/mixing of different substances in the pocket. The methods for our crystal structure analysis are clearly described in the Materials and Methods section (Lines 571-580).

9. Line 207. Please change 'is known' to 'is believed' or include a reference for this statement: pneumoniae topoisomerase IV) is known to interact with the catalytic residue Y120 during the DNA cleavage-religation reaction.

References (43, 44) included at the end of this sentence.

Additional reference (not included in the manuscript) listed below supports the statement.

Reference:

Nicholls RA, Morgan H, Warren AJ, Ward SE, Long F, Murshudov GN, Sutormin D, Bax BD. How Do Gepotidacin and Zoliflodacin Stabilize DNA-Cleavage Complexes with Bacterial Type IIA Topoisomerases? 2. A Single Moving Metal Mechanism. *Int J Mol Sci.* 2024 Dec 24;26(1):33. doi: 10.3390/ijms26010033. PMID: 39795899; PMCID: PMC11720246.

10. Please change the word 'key' for the word 'three'

245 We examined the interactions of BWC0977 in 9KGT and identified key interacting amino acids in

Reason: Supplementary Figure 9 in Bax et al., 2010 gives more interacting residues with GSK299423. Figure 4 suggests an additional contact with compound. and Extended Data Fig 6. does not detail how ' the key residues for BWC0977 binding' were selected.

We referred to the only interactions observed in crystal structure as key interactions, same is now changed to three (Line 246).

Methods.

Line 470-471. Consider changing:

'STEB loading buffer [40% (w/v) sucrose, 100 mM Tris-HCl 471 (pH 8), 0.5 mg/mL bromophenol blue, and 1 mM EDTA]'

to: 'STEB loading buffer [40% (w/v) sucrose, 100 mM Tris-HCl 471 (pH 8), 1 mM EDTA and 0.5 mg/mL bromophenol blue] ' Reason STEB = sucrose, tris, EDTA, bromophenol blue.

STEB expansion is corrected in the revised manuscript as suggested by the reviewer.

Line 482. Correct English.

change ' The assay plate was vortexed followed by a brief centrifuged to separate the phases.' to: 'The assay plate was vortexed then briefly centrifuged to separate the phases.'

This sentence is modified to "The assay plate was vortexed and briefly centrifuged to separate the phases".

Line 586: Include reference for search model 6WAA reference 39) as the search model.

39 Skepper, Colin K., et al. "Topoisomerase inhibitors addressing fluoroquinolone resistance in Gram-negative bacteria." *Journal of Medicinal Chemistry* 63.14 (2020): 7773-7816.

Reference 39 is now ref#42 and is included as a reference for search model in the revised manuscript (Line 589).

Include GESAMT reference:

E.Krissinel (2012), Enhanced Fold Recognition using Efficient Short Fragment Clustering, *J. Mol. Biochem.*, 1(2) 76-85. and CCP4 reference:

The suggested references are included as 62 and 63.

Reference 62: E. Krissinel (2012), Enhanced Fold Recognition using Efficient Short Fragment Clustering, *J. Mol. Biochem.*, 1(2) 76-85.

Reference 63: Agirre, J. et al. The CCP4 suite: integrative software for macromolecular crystallography. *Acta Crystallogr. Sect. Struct. Biol.* 79, 449–461 (2023).

Reviewer #3 (Remarks to the Author):

The manuscript reports structural and biochemical insights into the mechanism of BWC0977, a novel bacterial topoisomerase inhibitor (NBTI) in clinical development. The authors provide high-resolution cocrystal structures of BWC0977 bound to *Klebsiella pneumoniae* topoisomerase IV, alongside in vitro biochemical and resistance assays. The dual-target inhibition profile (gyrase and topoisomerase IV) is interesting and reveals a differentiated mechanism from fluoroquinolones.

Searching for new antibacterials and elucidating the mode of action of new antibiotics is due to increasing bacterial resistance urgently needed; therefore the scope of this paper is very interesting. However, there are some issues that need to be addressed prior to considering publication in *Nature Communications Biology*.

- Line 60 – please check the sentence “Despite the evolving resistance to these targets, the essential roles and conserved nature of DNA gyrase and topoisomerase IV make them promising candidates for new antibiotics.” Does resistance really evolve to the target or do the authors mean the drugs (fluoroquinolones).

This sentence is modified to “Despite the evolving resistance to existing drugs, the essential roles and conserved nature of DNA gyrase and topoisomerase IV make them promising candidates for new antibiotics”.

- Can you please update the introduction, as gepotidacin has been approved in March for treating uncomplicated UTIs caused by *E. coli*.

Gepotidacin FDA approval information mentioned and updated in Lines 75-77.

- Line 95 – correct the citation

This citation is corrected and included in the reference list as ref#37.

- Include standard deviation in the Table 1

Standard deviations are included in the Table 1.

- We suggest to perform also the multi step resistance studies, with serial sub-MIC passages to evaluate how fast the resistance can evolve over time (passages). Please compare the data with the positive controls (gepotidacin and ciprofloxacin).

We thank the reviewer for this suggestion. In fact, we had conducted studies for evaluating multi-step resistance development using serial sub-MIC passages for both gepotidacin and BWC0977. The results are presented in the figure below. The *gyrA*, *gyrB*, *parC*, and *parE*

genes were sequenced from colonies isolated on agar plates on Day 12; however, no mutations were detected in these genes.

Figure. Resistant mutant generation by serial passage method in *E. coli* ATCC 25922.

In contrast, earlier reported studies have clearly demonstrated the development of high-level resistance following repeated exposure to ciprofloxacin using the serial passage method. Upon stepwise exposure of quinolone-susceptible *E. coli* to increasing concentrations of ciprofloxacin (CIP), either in liquid or solid media, highly resistant mutants showing an 8,500- to 17,000-fold increase in fluoroquinolone resistance can be selected in vitro after as few as five serial passages¹. Similar findings have been reported in other Gram-negative bacteria, such as *Pseudomonas aeruginosa* and *Acinetobacter baumannii*².

1. Antimicrobial agents and Chemotherapy, June 1994, p. 1284-1291 Vol. 38, No. 6. Characterization of Fluoroquinolone-Resistant Mutants of *Escherichia coli* Selected In Vitro. Peter Heisig and Regina Tschorny
2. J Antimicrob Chemother 2012; 67: 2665–2672 doi:10.1093/jac/dks276 Advance Access publication 16 August 2012. In vitro evaluation of the potential for resistance development to ceragenin CSA-13. Jake E. Pollard, Jason Snarr, Vinod Chaudhary, Jacob D. Jennings, Hannah Shaw, Bobbie Christiansen, Jonathan Wright, Wenyi Jia, Russell E. Bishop and Paul B. Savage

- In the literature has been proven that the MICs for NBTIs are substantially more potent in knock-out strains lacking efflux pumps and bacterial strains with increased permeability. We suggest to perform the MICs also against mutant strains to prove that efflux pump or permeability is not the major issue with this compound.

We observe a 10-fold difference in the MICs of BWC0977 (0.001 µg/ml vs 0.01 µg/ml) between the *E. coli* Δ *tolC* strain, which lacks the efflux pump and the wild-type strain. However, given

that BWC0977 exhibits potent antibacterial activity against majority of global drug- resistant clinical isolates, with many likely to be harboring multiple efflux pumps and reduced permeability, this may not be of serious concern.

- Please clarify if all the enzyme assays kits have been obtained from Inspiralis or only the DNA gyrase and topoisomerase IV cleavage-complex assays.

All enzyme assay kits were obtained from Inspiralis (includes all gyrase supercoiling, topoisomerase IV decatenation and cleavage assay kits).

Rebuttal

We are pleased that Reviewers 1 and 3 have recommended the manuscript for publication and that Reviewer 2 acknowledged the successful rebuttal of their comments. We have carefully considered all suggestions and revised the manuscript accordingly to improve clarity, accuracy, and completeness.

Below, we summarize how each point raised by Reviewer 2 has been addressed in the revised manuscript.

Reviewer #1 (Remarks to the Author):

I can confirm that the authors have satisfactorily addressed my comments.

Reviewer #2 (Remarks to the Author):

Report on rebuttal for:

'Structural and biochemical insights into the mechanism of BWC0977 - a clinical stage broad-spectrum novel bacterial topoisomerase inhibitor'.

The **authors have successfully rebutted the reviewers comments**. The compound is undoubtedly of major interest, and this article deserves publication.

However, I would recommend, before the article is published one or two minor changes to improve its clarity. The literature is full of wrong structures, many of which are at limited resolution and the authors do not seem to fully understand which structures are incorrect.

Author response: We agree that clarity regarding structural reliability is important. All structures used for modelling and comparison were chosen based on the highest available resolution, and our comparative analysis highlights both positional differences and similarities in the binding of NBTI and quinolone-like compounds. We are confident that the resolution of these structures does not compromise the validity of our conclusions.

Also, GSK, have created some confusion in the literature with the name NBTI. BWC0977 is an NBTI, similar to gepotidacin which is also an NBTI. However, zoliflodacin is not an NBTI and binds at the same site as fluoroquinolones (as the authors correctly state). The paper would be improved by separating out these two novel classes of antibacterials (lines 68-81) targeting DNA Gyrase and topoisomerase IV. The NBTI gepotidacin was approved by the FDA in March 2025. Zoliflodacin, which is due to go to the FDA in December 2025, has very different safety barriers to overcome before approval. hERG safety is not, to the best of my knowledge, an issue with zoliflodacin.

Author response: We thank the reviewer for recognising that we correctly state that zoliflodacin is a not an NBTI. We have further modified the said sentence to reflect the non NBTI nature of zoliflodacin.

Please also note a missing reference:

Blower, Tim R., Benjamin H. Williamson, Robert J. Kerns, and James M. Berger. "Crystal structure and stability of gyrase–fluoroquinolone cleaved complexes from Mycobacterium tuberculosis." Proceedings of the National Academy of Sciences 113, no. 7 (2016): 1706-1713.

Author response: We have included the suggested reference.

And please change Line 54: topoisomerase IV, resulting in less effective fluoroquinolone binding^{12,13}. To both include the new reference and to acknowledge Wohlkonig et al., 2010. So it should read something like: topoisomerase IV, resulting in less effective fluoroquinolone binding¹¹⁻¹⁴. where reference 14 = Blower et al., 2016 (see above).

Author response: We have included the suggested reference as ref# 14.

If the authors care about accuracy perhaps they could also say: 'the first published structure'.

I understand GSK have unpublished higher resolution NBTI structures with a gram negative topo IV.

Author response: We are not privy to GSK's unpublished data.

Reviewer #3 (Remarks to the Author):

The authors have thoroughly addressed all the concerns raised in the first round of review. The updated introduction, corrections to citations and data presentation, as well as the additional experiments suggested, have substantially strengthened the manuscript. I recommend acceptance of the manuscript in its present form.

We hope that the revisions and clarifications provided in the manuscript and this rebuttal letter adequately address the reviewers' concerns. We are grateful for their insightful comments, which have helped improve the clarity and rigor of our work.

Thank you for considering our revised manuscript for publication.

Sincerely,
Nainesh Katagihallimath